# Plume detection and estimate emissions for biomass burning plumes from TROPOMI Carbon monoxide observations using APE v1.1

Manu Goudar[1], Juliëtte C. S. Anema[2], Rajesh Kumar[3], Tobias Borsdorff[1], and Jochen Landgraf[1]

[1]SRON Netherlands Institute for Space Research, Leiden, The Netherlands
[2]Royal Netherlands Meteorological Institute (KNMI), The Netherlands
[3]National Center for Atmospheric Research (NCAR), USA

**Correspondence:** Manu Goudar (manugv@sron.nl)

**Abstract.** This paper presents the Automated Plume Detection and Emission Estimation Algorithm (APE), developed to detect CO plumes from isolated biomass burning events and to quantify the corresponding CO emission rate. APE uses the CO product of the Tropospheric Monitoring Instrument (TROPOMI) aboard the Copernicus Sentinel-5 Precursor (S5P) satellite, launched in 2017 and collocated active fire data from the Visible Infrared Imaging Radiometer Suite (VIIRS), the latter flying 3 min ahead of S-5P. After identifying appropriate fire events using VIIRS data, an automated plume detection algorithm based on traditional image processing algorithms selects plumes for further data interpretation. The approach is based on thresholds optimised for data over the United States in September 2020. Subsequently, the CO emission rate is estimated using the cross-sectional flux method, which requires horizontal wind fields at the plume height. Three different plume heights were considered and the ECMWF reanalysis v5 data (ERA5) was used to compute emissions. A varying plume height in the downwind direction based on 3D Lagrangian simulation was considered appropriate. APE is verified for observations over Australia and Siberia. For all fire sources identified by VIIRS, only 16 % of the data corresponded to clear sky TROPOMI CO data with plume signature. Furthermore, the quality filters of APE resulted in emission estimations for 26 % of the TROPOMI CO data with plume signatures. Visual filtering of the APE's output showed a true-positive confidence level of 97.7 %. Finally, we provide an estimate of the emission uncertainties. The greatest contribution of error comes from the uncertainty in GFAS injection height that leads to emission errors < 100 %, followed by systematic errors in the ERA5 wind data. The assumption of constant emission during plume formation, and spatial under-sampling of CO column concentration by TROPOMI, yield an error of < 20 %. The randomised errors from the ensemble ERA5 wind data are found to be less than 20 % for 97 % of the cases.

## 1 Introduction

Carbon monoxide (CO) is an air pollutant and in high concentrations harms human health. It is an indirect greenhouse gas as it alters atmospheric OH, thus leading to an increase in the lifetime of methane (Spivakovsky et al., 2000). CO is produced mainly by incomplete combustion and Andreae et al. (1988); Watson et al. (1990) showed that biomass burning is a significant source of atmospheric CO. Furthermore, Granier et al. (2011); Crippa et al. (2018); Hoesly et al. (2018) showed an increase in CO emissions due to fossil fuel burning since 2000. CO emitted by localised sources on the ground leads to a prominent footprint in the atmosphere due to its lifetime ranging from days to several months (Holloway et al., 2000). These atmospheric

characteristics can be observed by satellites, which can provide essential information to improve our understanding of the effect of CO on air quality and climate.

The Tropospheric Monitoring Instrument (TROPOMI) aboard the Sentinel-5 Precursor (S5P) satellite, launched in 2017, monitors CO daily on a global scale (Borsdorff et al., 2018) and with a high spatial resolution of $7 \times 7 \ \mathrm{km}^2$, improved to $5.5 \times 7 \ \mathrm{km}^2$ in August 2019. Due to the high spatial resolution and daily coverage, CO emissions from cities (Borsdorff et al., 2019a, 2020; Lama et al., 2020), forest fires (Schneising et al., 2020; Li et al., 2020; Magro et al., 2021; van der Velde et al., 2021) and industrial sources (Tian et al., 2021) have been investigated and quantified. Rowe et al. (2022) compared TROPOMI CO measurements with in-situ aircraft measurements for different biomass burning plumes in 2018 and found differences of < 7.2 %, illustrating the potential of TROPOMI CO measurements to quantify CO emissions from these sources.

Most of the studies mentioned above estimated CO fluxes on large regional scales (Schneising et al., 2020; Magro et al., 2021; van der Velde et al., 2021) and mega-city scales (Borsdorff et al., 2019a, 2020; Lama et al., 2020). So far, the single-point CO emissions estimated from the TROPOMI CO data have received less attention. Tian et al. (2021) showed CO emissions based on TROPOMI for single-point industrial sources from India and China. They performed a statistical study for three years, as the geo-location of the industrial source is known. A similar analysis to quantify emissions from single-point biomass burning (fires) using TROPOMI CO data has not been shown in the literature.

Fire locations can be detected using the Visible Infrared Imaging Radiometer Suite (VIIRS) 375m thermal anomalies/active fire product (Schroeder et al., 2014). The VIIRS instrument is aboard the joint NASA/NOAA Suomi National Polar-orbiting Partnership (Suomi NPP) satellite and flies in the same orbit as S5P, in loose formation with a temporal separation of 3.5 minutes between them. This short time difference allows us to collocate observations of TROPOMI CO data and the VIIRS active fire product.

Different methods of estimating emissions are discussed in the literature, namely, inversion methods coupled with Gaussian dispersion models (Krings et al., 2011; Nassar et al., 2017; Lee et al., 2019), different Chemical Transport Models (CTM) (Brasseur and Jacob, 2017), Cross-sectional Flux Methods (CFM) (White et al., 1976; Beirle et al., 2011; Cambaliza et al., 2014, 2015; Kuhlmann et al., 2020) and integrated mass enhancement (IME) method (Frankenberg et al., 2016). An inversion coupled with a Gaussian plume model is used for flux inversions of an isolated single plume assuming steady and uniform wind conditions. This method fits an analytically computed Gaussian plume to TROPOMI CO column observations and can only be applied to observations under specific wind conditions (Varon et al., 2018). The IME method relates the emission and the integrated mass in the observed plume, and Frankenberg et al. (2016) showed that the relation is linear based on aircraft data for methane plumes. However, no such relationship has been established for CO measurements around fires. Hence, IME is not considered for the present work. Inversion methods using CTMs, such as, the Weather Research and Forecasting model coupled to Chemistry (WRF-Chem) (Grell et al., 2005), GEOS-Chem (Bey et al., 2001), and others, can reduce uncertainties and thus predict emissions more accurately. Although these methods can be applied to complex emission events, the corresponding simulations are complex, computationally expensive and difficult to automate, particularly for a large number of fires with different geo-locations which is the objective of this study. The CFM is well suited to the present work, as it requires less computational power and is easier to automate. CFM is based on the mass conservation of the pollutant transport in the

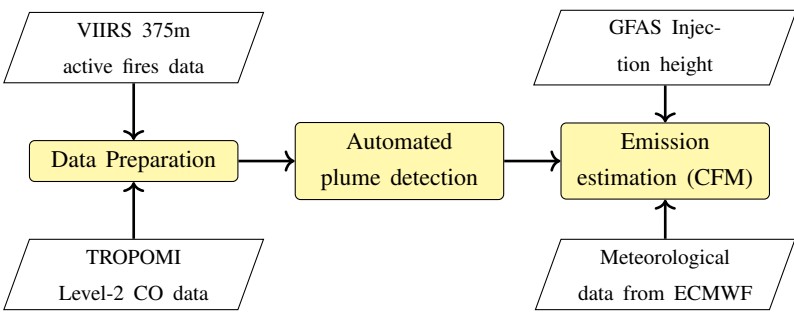

**Figure 1.** High-level flow chart of the APE algorithm.

downwind direction of the plume. The emission is estimated from corresponding fluxes across different planes perpendicular to the direction of the plume using the wind velocity at the plume height. Brunner et al. (2019) showed that the plume height depends on different aspects, namely, meteorology, emission height, etc., and may not be explicitly available. Furthermore, the CFM breaks down when diffusion is dominant, that is when the wind velocity is $< 2 \, \mathrm{ms^{-1}}$ (Varon et al., 2018).

The present work aims at developing an automated scheme to detect single and spatially isolated emissions from biomass burning events in TROPOMI observations and to estimate the corresponding CO emissions. For this purpose, we employed and improved the CFM. First, VIIRS fire data and satellite data were prepared for automated plume detection which is discussed in Sect. 2.1. The plume detection algorithm from a single point source using VIIRS fire counts is the subject of Sect. 2.2. Section 2.3 describes the emission estimation using the CFM where an appropriate choice of the plume height and the wind fields is discussed. The results of our study are discussed in Sect. 3, and finally, Section 4 concludes our study and sets recommendations for future work.

## 2 Methodology

Figure 1 illustrates a high-level flow diagram of the automated plume detection and emission estimation algorithm (APE), and the corresponding pseudo-code is given in Appendix B Algorithm 1. APE is divided into three parts, namely data preparation, automatic plume detection, and emission estimation. The data preparation algorithm identifies single-point fire sources from the VIIRS 375 m active fire data product (Schroeder et al., 2014) and subsequently selects and extracts TROPOMI CO data around every located fire source. Thereafter, the plume detection algorithm searches for a plume in the extracted CO data, and a detected plume serves as an input for emission estimation. The emission estimation algorithm initially computes the background CO, which is the usual observed CO concentration at that location without any CO emissions from the fire. The background allows us to obtain the CO enhancement, which is used by the CFM to estimate CO emissions. These three parts of the algorithm are discussed in detail in the following sections.

## 2.1 Data preparation

### 2.1.1 Selection of fire events

Fire events are inferred from the VIIRS 375 m active fire data product (Schroeder et al., 2014), provided by the Fire Informa-
tion for Resource Management System (FIRMS). FIRMS is operated by NASA's Earth Science Data and Information System
(https://earthdata.nasa.gov/active-fire-data). The data include various parameters such as fire radiative power (FRP), tempera-
ture and the time of measurement defined in latitude-longitude coordinates. Each of these coordinates corresponds to the centre
of a $375 \times 375 \ \mathrm{m}^2$ ground pixel and is referred to in this paper as a fire count or a VIIRS pixel. In most cases, a fire within a
single VIIRS pixel cannot create a CO signature spanning multiple TROPOMI pixels due to the detection limit of the satellite.
Only larger fires with a cluster of VIIRS fire counts can lead to a detectable CO plume in the TROPOMI observations. We
used the Density-Based Spatial Clustering of Applications with Noise (DBSCAN) algorithm (Ester et al., 1996; Schubert et al.,
2017) from the scikit-learn library (Pedregosa et al., 2011) to identify fire clusters. It separates areas that are densely packed
with fire counts from areas of low density and therefore has the ability to detect clusters of any shape. DBSCAN takes two
inputs; the first is the maximum search radius $r_{max}$ around a fire count, and the second is the minimum number of fire counts
within the area $n_{\min}$. $r_{max}$ is set to 4 km which is about half the size of the TROPOMI pixel. The minimum number of fire
counts has been empirically set to $n_{\min} = 10$. For further analysis, we converted each cluster into a single point source using
the fire radiative power (FRP) as the weight of the individual fire counts. This single-point source will henceforth be referred
to as a fire source and will be used as input to the TROPOMI CO data preparation.

### 2.1.2 TROPOMI CO data preparation

For the VIIRS fire sources, the corresponding TROPOMI orbits (see Table B1 for the version of the L2 product) were selected
and the orbit was corrected for stripes (see the Fast Fourier Transformation algorithm of Borsdorff et al. (2019b)). Figure 2
shows an example of the collocated information for a part of a TROPOMI orbit over Australia. Then, we extracted a data
granule of $41 \times 41$ TROPOMI CO pixels centred on each fire source. The minimum granule size of $220 \ \mathrm{km}$ was chosen, as an
air mass with an average velocity of $5 \ \mathrm{ms}^{-1}$ takes 6 h to reach the edges of the granules from the centre. After extraction, two
data quality filters are applied.

DP-1 The maximum TROPOMI CO pixel size due to distortion in the swath direction is restricted to $< 12$ km to avoid large
pixel size and its variation within the granule.

DP-2 For a data granule, 80 % of all CO data must meet a data quality $q_a > 0.5$ (Apituley et al., 2018), which corresponds
to clear-sky, clear-sky-like and mid-level cloud observations. Furthermore, we require 85 % of the pixels in an area of
$7 \times 7$ pixels centred on the fire source to meet the above criterion. The more usable pixels around the source, the better
the plume can be disentangled from the atmospheric background (see discussion in Sect. 2.3.1).

The threshold values are empirically determined for a reference data set from September 2020 over the United States and verified for two other data sets over Australia and Siberia (see the Tab. B1 in Appendix B for a detailed specification of the data sets). Finally, the selected CO scene is passed on as input to the plume detection algorithm.

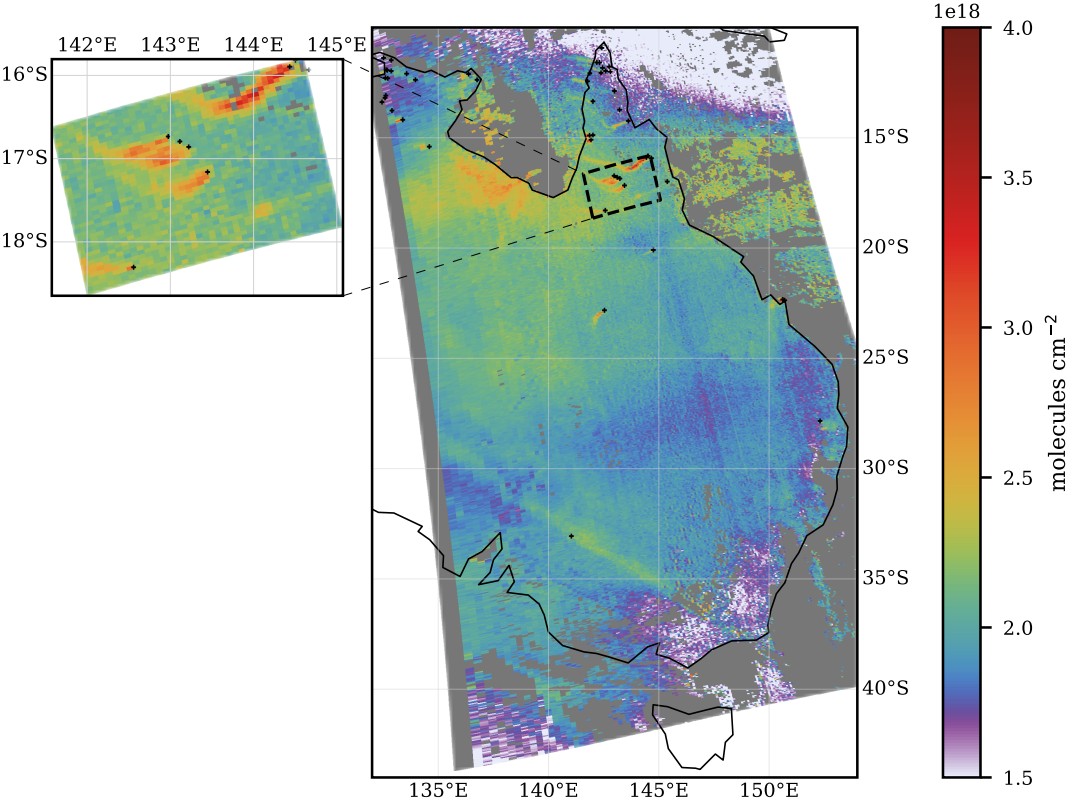

**Figure 2.** 49 detected fire sources represented by black '+' on 6 October 2019 overlapped with the TROPOMI level 2 CO data for orbit 10254. The dashed region represents a $41 \times 41$ pixel granule.

## 2.2 Plume detection algorithm

The next step in APE is to identify the plumes within each selected CO data granule. Kuhlmann et al. (2019) developed a plume detection algorithm based on statistical methods, and Finch et al. (2021) used machine learning to detect plumes. In the present study, a machine learning approach is not considered mainly due to the unavailability of data containing detected plumes and their sources for training. Instead, our plume detection approach is based on traditional image processing algorithms (van der Walt et al., 2014).

Using the extracted CO TROPOMI data, a plume is detected by a region-based segmentation algorithm, where pixels with similar properties are clustered together to form a homogeneous region. One of the most commonly used and classic region-based segmentation algorithms is the 'marker-based watershed transform method' (Beare, 2006; Gao et al., 2004). The CO

column concentration metaphorically represents the altitude of a topographic map. Thus, the watershed algorithm segments the regions into valleys and mountains (CO enhancements) based on a given marker and a gradient map. In the following paragraphs, we describe plume detection in more detail using an example.

The marker-based watershed algorithm in the scikit-image package (van der Walt et al., 2014) takes two inputs to segment an image. One is the 'gradient map' $\mathbf{I}_{\mathrm{grad}}$, which emphasises changes in altitude and attenuates homogeneous regions. The second input is a marker image $\mathbf{I}_{\mathrm{mark}}$ that provides the seed points for the algorithm, referenced by an integer label.

We start with the extracted CO TROPOMI granule of size $41 \times 41$ pixels $\mathbf{I}(i,j)$ with $i,j = 1, \cdots, 41$. An example is shown in Figure 3(a). First, the high-frequency components of the CO-image are reduced by a 2D Gaussian filter with a standard deviation of $\sigma = 0.5$ pixels, chosen empirically. The smoothed image is called $\mathbf{I}_{\mathrm{s}}$. From this image, the gradient map $\mathbf{I}_{\mathrm{grad}}$ is computed using a Sobel operator (Sobel and Feldman, 2015; van der Walt et al., 2014), namely

$$\mathbf{I}_{\mathrm{grad}} = \sqrt{\mathbf{G}_x + \mathbf{G}_y} \tag{1}$$

with

$$\mathbf{G}_x = \begin{bmatrix} 1 & 0 & -1 \\ 2 & 0 & -2 \\ 1 & 0 & -1 \end{bmatrix} * \mathbf{I}_{\mathrm{s}} \qquad \mathbf{G}_y = \begin{bmatrix} 1 & 2 & 1 \\ 0 & 0 & 0 \\ -1 & -2 & -1 \end{bmatrix} * \mathbf{I}_{\mathrm{s}}, \tag{2}$$

where $*$ represents the convolution operator. Here, the gradient $\mathbf{I}_{\mathrm{grad}}$ emphasises the edges of a plume, as shown in Fig. 3(b).

By default, the marker image ($\mathbf{I}_{\mathrm{mark}}$) is initialised to zero and then two different seeds are defined. One seed indicates regions without CO enhancements and another refers to regions with clear CO enhancements given by $\mathbf{I}_{\mathrm{mark}}(i,j) = 1$ and $\mathbf{I}_{\mathrm{mark}}(i,j) = 2$, respectively. The seeds are defined as follows:

1. The regions without CO enhancement: A pixel $\mathbf{I}_{\mathrm{mark}}(i,j)$ does not have CO enhancement if it is below the median of $\mathbf{I}_{\mathrm{S}}$ or below the mean of the $15 \times 15$ pixels centred at $\mathbf{I}_{\mathrm{S}}(i,j)$. The size of $15 \times 15$ pixels was chosen to account for the background variability. Pixels corresponding to no enhancement can be seen in Fig. 3(c) represented by the label '1' and the image is referred to as a preliminary marker image.

2. The regions of CO enhancement: Using the preliminary marker image with labels '0' and '1', we identified all connected pixels with the same marker value (hereafter referred to as connected regions) using the 'label' algorithm (Fiorio and Gustedt, 1996) of the scikit-image package (van der Walt et al., 2014). Each connected region is identified by a unique integer value per pixel (not to be confused with the seed marker). Next, we zoomed in on a $5 \times 5$ pixel area around the fire source and extracted all connected regions as potential plumes. Further, the potential plumes were expanded by going to $15 \times 15$ pixels around the fire source using pixels with the same label. Then, the labelled CO data in this area were extracted and a CO threshold was calculated as their mean value. Lastly, all pixels within the $15 \times 15$ pixel area were marked with a label '2', if their CO value was above this threshold. This yields the remaining seed points which are defined in a $15 \times 15$ pixel area around the fire source.

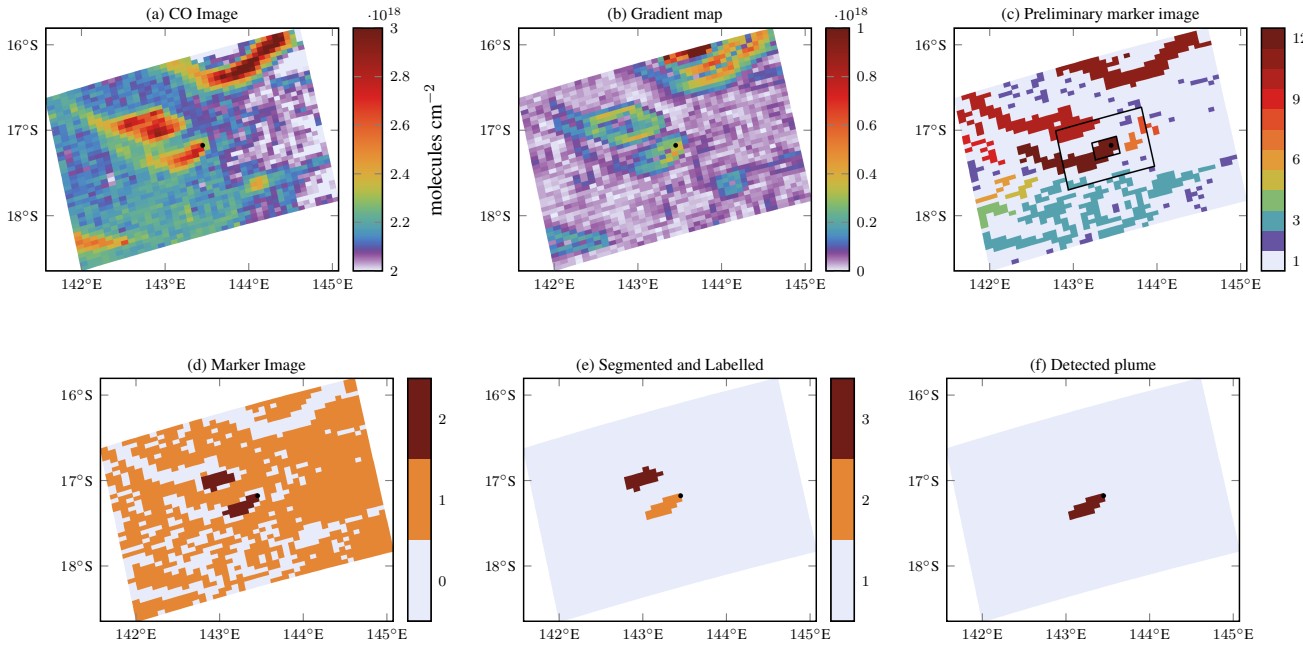

**Figure 3.** Plume detection algorithm. An example showing different steps for a fire source on 6 October 2019 in Australia.

The above selection process is illustrated in Fig. 3(c) with the different labels of the connected regions. The final marker image is shown in Fig. 3(d). Finally, the watershed algorithm computes a segmented image for the entire domain using the gradient map image $\mathbf{I}_{\mathrm{grad}}$ and the marker image $\mathbf{I}_{\mathrm{mark}}$. Figure 3(e) shows an example of a segmented image. Using the gradient map, the watershed algorithm has decided that the two areas of enhanced CO values are not connected, and therefore do not belong to the same plume. From the segmented image, we extracted the correct plume, which should originate from emissions at the source location. Therefore, we consider only those labelled areas that overlap with the centre $7 \times 7$ pixels. Figure 3(f) shows the detected plume. The detected plume appears to be shorter in this case, but the tail end of the plume, i.e., around $< 143^0$ E, will fail background subtraction due to similar enhancements as the background. This can also be seen in the gradient map, where no gradient is detected at the top side of the plume.

Finally, the suitability of the extracted plume for further processing is evaluated and the length of the plume is calculated. The plume is provided to the emission estimation module if:

PD-1  The plume length is $> 25$ km.

PD-2  If there are not more than nine non-clustered fire counts or any other identified fire cluster within $0.05°$ distance from or in the identified plume.

If the length of the plume is $< 25$ km, then the detected plume is flagged as a short plume and will be ignored for further processing. The short plumes are difficult to quantify in an automated way as they can have different shapes, which makes it

difficult to identify the direction of the plume. The second criterion (PD-2) removes all plumes with multiple fire sources, as the aim of this paper is to quantify fires with single sources.

## 2.3 Emission estimation

For detected plumes, emissions were estimated using the cross-sectional flux method (CFM) (White et al., 1976; Beirle et al., 2011; Cambaliza et al., 2014, 2015; Kuhlmann et al., 2020). The CO emission $E$ is defined as the mean flux through $n$ cross sections perpendicular to the downwind direction of the plume, namely

$$E = \frac{1}{n} \sum_{i=1}^{n} Q_i$$

$$Q_i = \int \delta C_{\mathrm{co}}^i(s, t_0) \cdot v^i(z_i, s, t_0) \cdot ds \tag{3}$$

where $Q_i$ (in $\mathrm{kgs}^{-1}$) is the CO flux through cross-section $i$, $\delta C_{\mathrm{co}}^i$ (in $\mathrm{kgm}^{-2}$) is the background subtracted CO values along a cross-section $i$ and $v^i$ (in $\mathrm{ms}^{-1}$) is the velocity perpendicular to the cross-section $i$. Wind velocity $v(z, s, t_0)$ at plume height $z$, cross-section position $s$ and observation time $t_0$ are obtained from the data of the European Centre for Medium-Range Weather Forecasts Reanalysis v5 (ERA5) (Hersbach et al., 2017). For error characterisation, we define the standard error ($\sigma_{\mathrm{E}}$) as

$$\sigma_{\mathrm{E}} = \frac{1}{n} \sqrt{\sum_{i=1}^{n} (E - Q_i)^2} \tag{4}$$

The cross-sections, hereafter referred to as transects, were determined by calculating the plume direction in the downwind direction. The plume line results from a second-order curve fit through the pixel centres of the detected plume (see, e.g. the black solid line in Fig. 4(a)). Next, the transects at every 2.5 km perpendicular to the plume line were calculated and are illustrated as dashed lines in Fig. 4(a). The transects are sampled at 2.5 km to reduce errors due to interpolation, discussed in the next paragraph. To compute $Q_i$ in Eq. 3, each transect was sampled at distances of 500 m. Points over transects are over-sampled to obtain a smoother CO distribution, which further helps in the background subtraction discussed in Sect. 2.3.1. Along each transect, the CO column ($C_{\mathrm{co}}$) is extracted by linear interpolation of the original CO data and is illustrated by a dotted black line in Fig. 4(b). This CO column is further used to calculate $\delta C_{\mathrm{co}}$ in Eq. 3. During the diagnostic tests of our interpolation algorithm, an oscillation was observed in the CO columns integrated along the transects as a function of the downward distance from the fire source (see Fig. 4(c)). The oscillation is due to the under-sampling of the CO distribution by the TROPOMI instrument. The distance between two minima is approximately equal to the TROPOMI pixel size. This error propagated further into the CO enhancement $\delta C_{\mathrm{CO}}$, which was computed from the background subtraction algorithm.

### 2.3.1 Background Subtraction

To determine the atmospheric background of CO per transect, first, we re-centre the $C_{\mathrm{CO}}$ such that the maximum is at the origin to facilitate the Gaussian fit. The transect line is truncated at the first minima of CO on either side of the origin, as illustrated by the red line in Fig. 4(b). To determine the background for each transect (red line), we assume that the column CO

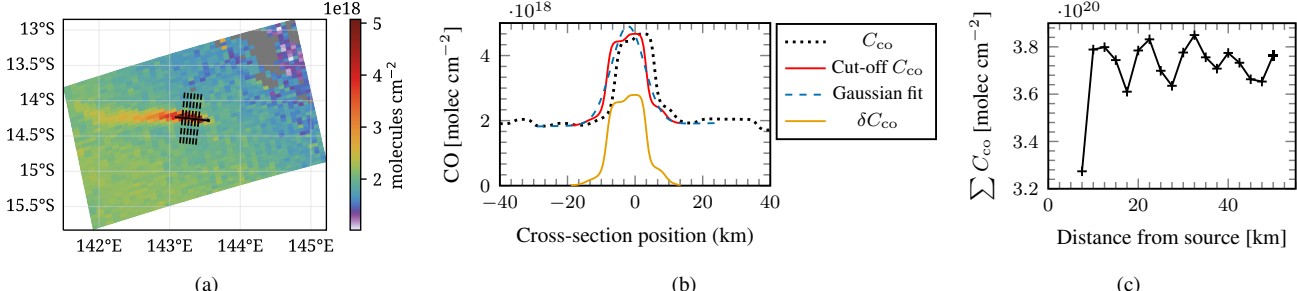

**Figure 4.** Plume on 1 October 2019 at 03:52 UTC. (a) Plume and every second transact lines drawn based on the detected plume separated by 5 km in downwind direction. (b) The black dotted line corresponds to the CO column along a transect in (a), and the red line shows re-centred and cut-off CO used for Gaussian fitting. The blue dash-dotted line corresponds to the Gaussian fit, and the orange line represents the enhanced CO along the transect. (c) $\sum C_{\mathrm{co}}$ along a transect against the distance from the source.

along the transect can be expressed as

$$C_{\mathrm{co}} = H_0 + H_1 \cdot s + A_0 G(s) \tag{5}$$

where $H_0$ and $H_1$ represent the background and the slope of change in the background over the transect, respectively. $A_0$ is the amplitude of the Gaussian distribution $(G)$. We determined the background by fitting Eq. 5 through the CO data, which is subsequently subtracted from the $C_{\mathrm{co}}$ data to calculate the CO enhancement as shown below

$$\delta C_{\mathrm{co}} = \max\{0,\ C_{\mathrm{co}} - H_0 + H_1 \cdot s\}. \tag{6}$$

Here, the negative enhancements in the CO column are ignored. The blue-dashed and the orange lines in Fig. 4(b) represent Gaussian fit and $\delta C_{\mathrm{co}}$, respectively.

### 2.3.2 Filtering during background subtraction

The background subtraction includes an important filtering mechanism to remove overlapping plumes. This is done during the background subtraction after the transect line is truncated. The filter criterion is as follows

EE-1 The difference between the minima on either side of a truncated transect should be $< 10$ %. This ensures a smooth background and the absence of any interference with adjacent emission events.

### 2.3.3 Plume height

The plume height $z_i$ at a transect/cross-section $i$ is used to extract the appropriate wind velocity $v(z_i, s, t_0)$. For wildfires, Rémy et al. (2017) showed that the Integrated monitoring and modelling System for wild-land Fires (IS4FIRES) injection height, $z_{\mathrm{inj}}$, from the Global Fire Assimilation System (GFAS) database is in good agreement with the observations. Sofiev et al. (2012) showed the IS4FIRES injection height deviated by less than 500 m compared to the MISR Plume Height Project (MPHP),

therefore we consider 500 m as plume height uncertainty. First, we assume that the height of the plume is $z_{\mathrm{inj}}$ and is constant throughout the plume. This may be true for stable meteorological conditions. The constant plume height will be called $z_{\mathrm{c}}$ and the uncertainty at this plume height is given as $z_{\mathrm{c}}^p = z_{\mathrm{inj}} + 500$ m and $z_{\mathrm{c}}^m = z_{\mathrm{inj}} - 500$ m, respectively.

It should be noted that the injection height calculated in GFAS is for 24 h and may not be appropriate for a satellite plume which is a snapshot at a measurement time $t_0$. In addition, the plume height may vary due to meteorology in the downwind direction. Therefore, we alternatively simulated particle trajectories starting at the fire site around the injection height with a three-dimensional Lagrangian tracer dispersion model. The local plume height $z_i$ is then estimated by averaging the height of the tracers along the downwind direction. This estimated plume height is $z_{\mathrm{lag}}$ and captures the change in height in the downwind direction.

The Lagrangian simulations were performed using tracer particles. The motion of tracers is simulated according to

$$\frac{d\mathbf{x}_p(t)}{dt} = \mathbf{v}(\mathbf{x}_p(t)) \tag{7}$$

where $\mathbf{v}(\mathbf{x}_p)$ represents the fluid velocity at the instantaneous particle position $\mathbf{x}_p$. The explicit forward Euler scheme (Butcher, 2003, p. 45) was used to integrate the equation in time. The velocity on the right-hand side of the Equation. (7) is calculated by tri-linear interpolation of the ERA5 velocity fields. The fire counts described in Section 2.1.1 are used as source locations for Lagrangian simulations. Three tracer particles are released at $z_{\mathrm{inj}}$ and $z_{\mathrm{inj}} \pm 500$ m at each source location. The release at $z_{\mathrm{inj}} \pm 500$ m is used for uncertainty analysis. The end time of the simulations is the TROPOMI measurement time $t_0 \approx 13:30$ h local time (Veefkind et al., 2012), and the simulation starts at $t_0 - 6$ h. The particles are released from the source locations every 2 minutes. Figure 5(a) shows a simulation of the tracer particles for one plume. The white band indicates particles at the TROPOMI measurement time.

The contribution to fire emissions is low in the early morning, as shown in the ecosystem-specific diurnal cycles by Li et al. (2019). Therefore, we ignore trajectory simulations before $t_0 - 6$ h. Additionally, the process of heating due to fires is not accounted for in our Lagrangian simulation as we assume the ERA5 velocity fields contain some aspect of heating, as ERA5 assimilates skin surface temperatures.

In each transect, the heights of the tracer particles released at $z_{\mathrm{inj}}$ were extracted and the mean height, $z_{\mathrm{lag},i}$ was calculated. This is assumed to be constant along the transect. Figure 5(b) shows the height of the plume for different transects from the fire source that was used to calculate the velocity $\upsilon$, in Equation 3. The uncertainty in plume height is defined as $z_{\mathrm{lag}}^p$ and $z_{\mathrm{lag}}^m$ and was calculated from tracer particles that were released at heights $z_{\mathrm{inj}} + 500$ m and $z_{\mathrm{inj}} - 500$ m, respectively, and can be observed in Figure 5(b). Finally, the velocity, $\upsilon$, was used to compute emissions.

### 2.3.4 Filtering during Lagrangian simulations

Related to the Lagrangian simulation, we apply three filters:

EE-2 The injection height from GFAS must be available.

EE-3 If the simulated trajectories are not aligned in the direction of the plume, then the plume is rejected.

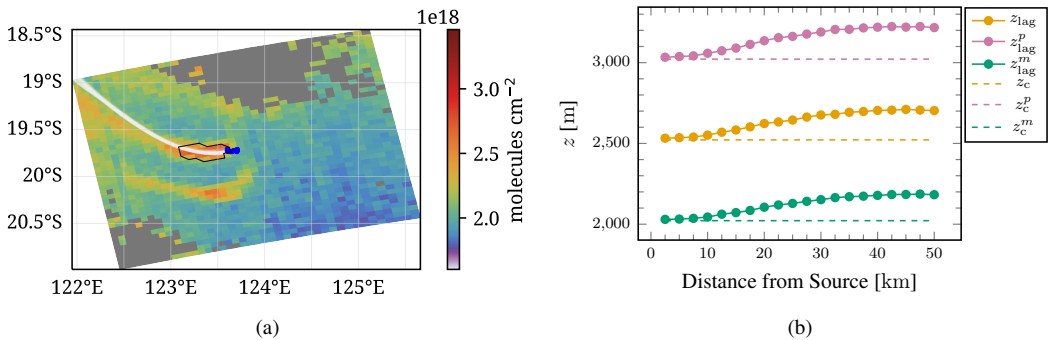

(a)  (b)

**Figure 5.** (a) The white band shows all tracer particles at the end of the Lagrangian simulation, and the blue dots show the fire counts on the detected plume. (b) Shows the plume height computed for different transects from Lagrangian simulations. The constant plume height ($z_{\mathrm{c}}$) represented by an orange dashed line is 2521.87 m.

EE-4  If the wind velocity at the TROPOMI measurement time used to compute emissions is less than $2\ \mathrm{ms}^{-1}$ then the plume is rejected.

Filter EE-2 may become relevant due to the false detection of a plume, false fire in the VIIRS active fire database or the missing data in the GFAS database. There are several potential origins for filter EE-3: the rotation, errors in ERA5 velocities, the spatial and temporal resolution of velocity fields or inaccurate injection height. Finally, if the wind speed is below the specified value in EE-4, diffusion dominates the pollutant transport, and CFM is not appropriate to estimate the CO emission.

## 3  Algorithm application

**Table 1.** Results for automated plume detection and emission estimation algorithm (APE v1.1) for four months in the US, Australia and Siberia

| Regions | Fire Clusters | CO data | Plume detection | Emission estimation | Visual Inspection |
|---------|---------------|---------|-----------------|---------------------|-------------------|
| US | 1081 | 213 | 130 | 37 | 35 |
| AU | 2013 | 385 | 266 | 129 | 128 |
| Sib Jun | 416 | 130 | 83 | 35 | 34 |
| Sib Jul | 2052 | 599 | 94 | 25 | 24 |
| All Regions | 5562 | 1327 | 378 | 226 | 221 |

The APE algorithm targets global performance and includes several threshold values, which need to be carefully determined for optimal performance. For the current version of APE V1.1, we decided to determine the thresholds using the region that encapsulates the United States of America (US) in September 2020. The algorithm is verified by applying it to other regions that encompass Australia (AU) in October 2019 and Siberia (Sib) in June and July 2021 (see Table B1 for more details). It is

important to note that these regions are not used to configure APE and can therefore be used to verify the overall performance of the algorithm. The different periods were chosen to focus on the regional burning season to maximise the number of fires observed. Table 1 shows the number of cases evaluated by different APE modules. The columns *Fire cluster* and *CO data*, *plume detection*, and *emission estimation* show the results for data preparation (see Sect. 2.1), plume detection (Sect. 2.2) and emission estimation (Sect. 2.3), respectively. Furthermore, the details corresponding to filtering can be found in Tables B2, B3 and B4 in the Appendix.

A total of 5562 fire sources (see Table 1) were identified in the VIIRS active fire data product for all regions based on the clustering method discussed in Section 2.1.1. For each fire source, the TROPOMI CO data were filtered for maximum pixel size and quality (see Sect. 2.1.2). The TROPOMI pixel size filter (DP-1) rejected 1533 cases out of 5562 cases that mostly belonged to Australia and the United States. The quality of TROPOMI CO data (DP-2) was found to be insufficient for about 2553 cases in 5562 cases, mainly due to the presence of clouds. For the Siberian region on July 2021, more than 50 % of all fire clusters are flagged as bad-quality data for the same reason. Finally, the data preparation part yielded a total of 1327 good CO data granules for all regions for further processing.

The plume detection algorithm described in Sect. 2.2 identified a plume signature in 882 cases for all regions from 1327 good CO data cases available. A total of 445 cases were found not to have enhancements (see Table B3), meaning that the enhancement of CO from these fires was below the detection limit of TROPOMI. In 882 cases, only 378 cases were considered good as the PD-1 filter flagged 309 identified plumes as short as their plume length was < 25 km. Furthermore, the PD-2 filter identified a total of 195 cases where other fire sources and clusters were present in the detected plumes.

The emission estimation algorithm took 378 plumes as input and calculated emissions using the CFM for a total of 226 cases. Therefore, a total of 152 plumes were rejected by the EE-1, EE-2, EE-3, and EE-4 filters during emission estimation (see Table B4 for details). The EE-1 filter removed 29 cases due to overlap with other plumes. The injection height from the GFAS database was not available for 57 cases (EE-2 filter). In addition, the particle-plume alignment filter, EE-3, removed a total of 51 cases. This can be attributed to poor plume detection, inaccurate velocities, or injection heights. Finally, the velocity filter, EE-4, rejected a total of 15 cases.

As a final step, the 226 cases were verified by visual inspection. We can see a good performance of the algorithm, as only five of the 226 cases were wrongly identified. Fig. 6 shows three examples of bad cases. Thus, based on the above analysis, we can conclude that 97.7 % of the cases produced by the algorithm are good.

Overall, APE incorporates strict data filtering, which is mainly driven by the TROPOMI detection limit and data quality due to cloud coverage. However, this should not distract from the fact that TROPOMI is the first instrument that shows these emission features in satellite data. Secondly, the data yield is thinned out further by selecting data which are appropriate for the current APE inversion scheme, which are fire emission events by isolated single sources. Overall, we consider that there will be sufficient data yield for a new TROPOMI CO data product when applied to more than 6 years of global observations.

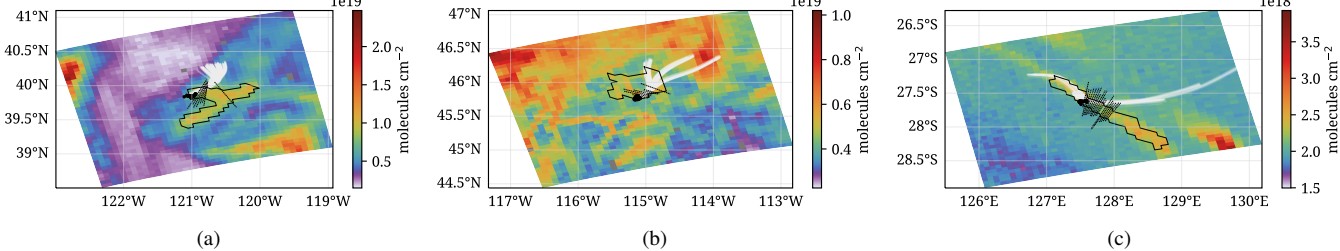

(a)    (b)    (c)

**Figure 6.** Falsely identified cases by the APE algorithm. Black dots indicate the fire counts, white bands the tracer particles and the black polygon depicts the detected plume, respectively. The black dashed lines are transaction lines.

## 3.1 Cross-Sectional Massflux method (CFM)

The CFM computed emissions for 221 cases. To compare the effect of plume heights, two variables were defined, namely, the mean plume height $\overline{z}_{\mathrm{lag}}$, which is the mean of $z_{\mathrm{lag}}$ of all transects along the downwind direction of the plume, and the maximum rise in plume height ($\delta z$) with respect to $z_c$. They are given as

$$\overline{z}_{\mathrm{lag}} = \frac{1}{m} \sum_{i=0}^{m} z_{\mathrm{lag},i} \tag{8}$$

$$\delta z = \max\{z_{\mathrm{lag}}\} - z_c \tag{9}$$

Figures 7 (a) and (c) show the mean plume height $\overline{z}_{\mathrm{lag}}$ plotted against the constant plume height ($z_c$) for the United States, Australia, and the Siberian region. $\delta z$ decreased and increased in the downwind direction for about 43 and 178 fires, respectively. Furthermore, $\delta z$ in the downwind direction was found to vary $> 500$ m for 30 fires in Australia and the United States, as shown by the orange colour in Fig. 7(a). However, no such cases were found in Siberia (see Fig. 7(c)). Among these 30 fires, about 11 fires had $\delta z > 1000$ m. This increase in plume height in the downwind direction can be attributed to the rising warm air, which may be heated by the fire. Furthermore, this heating can be related to total fire radiative power (FRP) and fire counts, since they describe the heat generated and the burnt area, respectively. However, no such relation was observed, as there were cases with low FRP, or low fire counts, where $\delta z > 1000$ m and vice versa. Additionally, it was challenging to find a suitable reason for a large increase in plume height in the downwind direction. Obviously, this plume height variation can influence the emissions due to the change in the velocity with height.

Figures 7(b) and (d) compare the emissions computed from the Lagrangian plume height ($E_{\mathrm{lag}}$) with the emissions computed from the constant plume height ($E_{\mathrm{c}}$) represented in black, and the 100 m plume height ($E_{100}$) represented in blue. A combination of all cases in Figs. 7(b) and (d) shows that $E_{\mathrm{c}}$ varied less than 10 % from $E_{\mathrm{lag}}$ for a total of 198 cases. For Siberia fires, the corresponding variation is less than 4 %. However, 23 cases in the United States and Australia show differences of $>10$ %. Thus, the overall effect of the Lagrangian plume height is considered minor. However, we could identify several cases where the emissions estimate from Lagrangian plume height becomes more reliable. For example, a US fire (black colour) on the bottom right of Fig. 7(b) was found to have a high $E_{\mathrm{lag}} = 809$ kgs$^{-1}$ and low $E_{\mathrm{c}} = 115.9$ kgs$^{-1}$. The total fire radiative

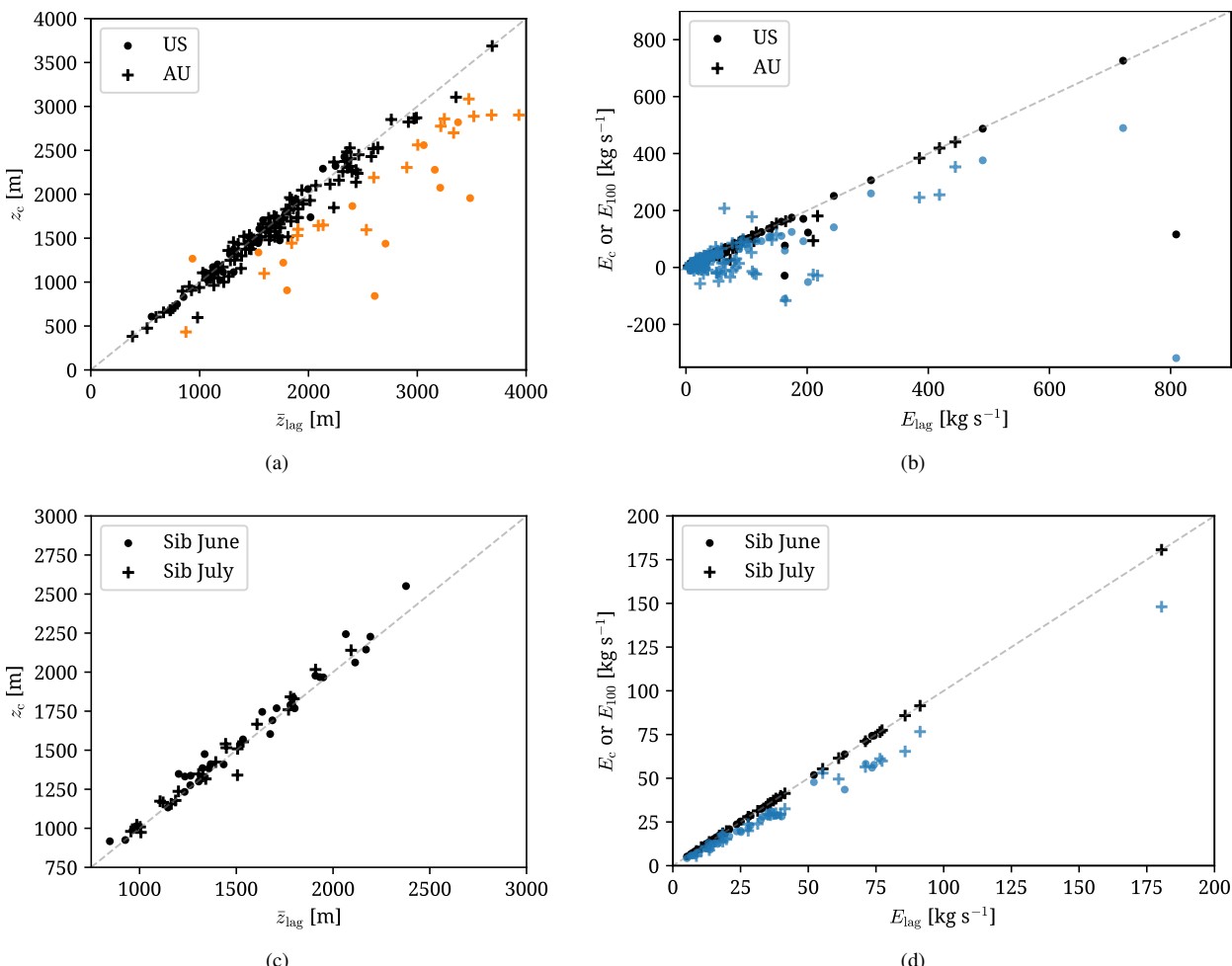

**Figure 7.** Plume height variation and emissions for regions encapsulating US and Australia (top figures) and Siberia (bottom figures). (Panel (a) and (c)) The mean plume height (see Eq. (8)) versus the constant plume height for each fire. The black colour represents $\delta z \leq 500$ m, and the orange colour indicates $\delta z > 500$ m (Panel (b) and (d)). Comparison between the emissions computed at plume height $z_{\text{lag}}$ versus $z_{\text{c}}$ represented by black colour and $z_{\text{lag}}$ versus a constant plume height of 100 m represented by blue colour.

power (FRP) for this case was the highest among all detected plumes, and the burnt area (number of fire counts in VIIRS data) was the third highest among all detected cases. The CO enhancement was also large, so a high emission estimate is expected. Furthermore, a high FRP is correlated with higher temperatures, so an increase in plume height in the downwind direction is normal. It should be noted that the Lagrangian simulations do not consider heating but we assume that the meteorological data (velocities) in ERA5 cover this, as it assimilates the surface temperatures. The increase in plume height is observed in Lagrangian simulations as $z_{\text{lag}}$ increases by 1350 m compared to the constant plume height in the downwind direction at 32.5

320  km from the fire source. From this, we can conclude that $E_{\text{lag}}$ may be more appropriate than $E_{\text{c}}$. Similar reasoning can be used to explain why $E_{\text{lag}}$ was higher compared to $E_{\text{c}}$, where the FRP on average was higher.

Figures 7(b) and (d) also compare emissions from Lagrangian plume height to a constant 100 m plume height. We considered 100 m plume height as three-dimensional velocity fields, which are required to compute the CO emissions based on the plume heights $z_{\text{c}}$ and $z_{\text{lag}}$, amounting to a large quantity of data. Furthermore, computing emissions by scaling 100 m winds (Hersbach

et al., 2018) would simplify the approach to a large extent. However, we found no correlation between the difference in the emissions ($E_{100} - E_{\text{lag}}$ or $E_{\text{c}}$) and the variation in plume heights. Additionally, a total of 37 fires were found to have negative values for $E_{100}$ due to a negative velocity at 100 m. This makes it challenging to find an appropriate scaling to obtain emissions at $z_{\text{lag}}$ from the velocities at 100 m, thus highlighting the importance of using three-dimensional velocity fields rather than surface-near-wind fields at a fixed altitude. From all these observations, we conclude that the varying plume height is more

reliable to compute emissions by an automated algorithm.

## 3.2 Emission uncertainty

Sherwin et al. (2023) validated satellite $CH_4$ data using controlled emission releases of point sources of methane for detection and quantification. For CO, such validation is not possible for single-point releases. (Rowe et al., 2022) have shown that the integrals of TROPOMI CO data along the plume transects were $\approx 7.2$ % higher than the aircraft measurements after

corrections for a few fires in the US. However, they do not report emissions. Thus, to the best of our knowledge, no independent, high-quality estimate of CO point-source emission is currently available. Hence, to demonstrate the data quality and enhance confidence in our data product, we instead perform a detailed uncertainty analysis disentangling (quasi-)random and systematic errors.

We estimate three different contributions to the uncertainty of the estimated emission, where constant emissions over time

are assumed for each fire case. First, the relative variation of the CO fluxes through the different intersects $Q_i$ is considered. Different error sources may cause this variation, and the corresponding error in the flux estimate can be characterised by the standard error $\sigma_{\text{E}}$ in Eq. 4. Second, errors due to random uncertainties in ERA5 velocity fields are addressed. Finally, systematic errors that affect the different fluxes $Q_i$ require a different approach. These errors are TROPOMI CO column biases, ERA5 velocity bias, and injection height uncertainty. To complete our error classification, we verify the emission uncertainty in the

APE algorithm using data from WRF simulations, where the wind velocities, CO and injection height are known.

### 3.2.1 Standard errors

The standard error of the emission estimate encompasses various uncertainty sources, e.g., the interpolation error due to the under-sampling of the CO field by TROPOMI (shown in Fig. 4(c)), the precision of the TROPOMI CO data, the uncertainty variation in defining the atmospheric CO background per intersect, and the temporal variation of the emission around its mean.

$\sigma_{\text{E}}$ does not allow us to disentangle these error sources, except for the TROPOMI CO precision, which is specified for every TROPOMI observation. Overall, the precision of the CO column is $< 10$ % per pixel, even for dark scenes over land (Landgraf et al., 2016). For the flux estimate, this yields a negligible error contribution. To compare the standard error for different fires,

**Table 2.** Maximum values of standard error and emission uncertainties due to plume height for different regions among all fires.

| Region | $\sigma_{\mathrm{E}}(z_{\mathrm{lag}})\%$ | $\sigma_{\mathrm{E}}(z_{\mathrm{c}})\%$ | $\Delta E_{\mathrm{lag}}^{p}\%$ | $\Delta E_{\mathrm{lag}}^{m}\%$ | $\Delta E_{c}^{p}\%$ | $\Delta E_{c}^{m}\%$ |
|--------|------|-------|-------|-------|--------|--------|
| US | 15.11 | 35.51 | 71.28 | 82.27 | 246.93 | 163.54 |
| AU | 18.79 | 28.10 | 94.78 | 95.41 | 130.17 | 170.37 |
| Sib Jun | 18.21 | 18.13 | 17.34 | 14.95 | 17.37 | 8.84 |
| Sib Jul | 19.72 | 19.57 | 14.88 | 13.12 | 15.47 | 12.1 |

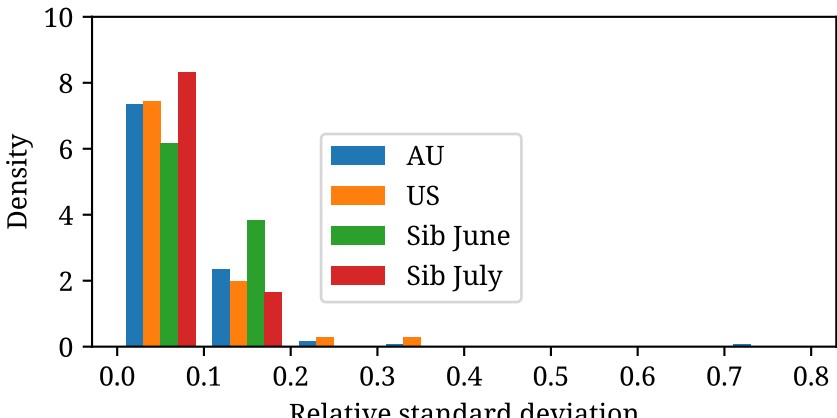

**Figure 8.** Histogram plot showing density vs relative standard deviation of emissions from 10 ERA5 ensemble velocity fields for four regions.

Tab. 2 reports the maximum relative standard error for the four regions using the Lagrangian plume height and the constant plume height $z_c$. The error for individual fires can be accessed in the database (Goudar et al., 2023). The data show that the

maximum standard error for the Lagrangian plume height is significantly smaller than for the constant plume height $z_c$ for both the US and Australia. This is another indication to use the Lagrangian plume height as a baseline for APE. For the Siberian region, there is no difference between the two methods because the height of the plume does not vary much as depicted in Fig. 7(c). In general, the standard error of the emission estimate is $< 20\,\%$.

### 3.2.2 ERA5 uncertainties

ERA5 ensemble data (Hersbach et al., 2017) is used to quantify velocity uncertainties. The ensemble includes ten members, and the variation between the members represents random errors, but not systematic errors (Hersbach et al., 2020). Due to the small size of the ensemble, the data cannot encapsulate all the random uncertainties. For every member $j$ of the ensemble, the Lagrangian plume height (Sect. 2.3.3 is used to calculate the emission $(E_{\mathrm{lag}}^{j})$ for all plumes. Subsequently, the relative standard deviation $\sigma_{\mathrm{vel}}/E_{\mathrm{lag}}$ per emission source was computed, where $\sigma_{\mathrm{vel}}$ represents the standard deviation of the emissions

of the ten members of the ensemble. Figure 8 shows the density histogram versus the relative standard deviation for all regions. A total of 215 cases among 221 cases have a velocity uncertainty of less than 20 %. One Australian case has an uncertainty

$> 70$ %, which was due to a single ensemble whose velocities were $\approx 3$ times higher than the other 9 ensembles. Although the ensembles do not fully describe the random errors, we observe less than 20 % uncertainty in 97 % of the cases.

### 3.2.3 Systematic errors

One potential error that cannot be addressed with the standard error is an overall bias in the TROPOMI CO product. Borsdorff et al. (2019b) reported a CO bias of 3.4 ppb for the TROPOMI product compared to the Total Carbon Column Observing Network (TCCON). This corresponds to a typical relative error $< 1.7$ % for a plume concentration of about 200 ppb in a plume. Rowe et al. (2022) showed that the integrals of TROPOMI CO data along the plume transects were $\approx 7.2$ % higher than the aircraft measurements after corrections for a few fires in the US. Assuming a worst-case scenario, the constant bias of 7.2 % over the plume leads to $\approx 7.2$ % higher emission estimation.

Another error in this category is the emission uncertainty due to the uncertainty in the IS4FIRES injection height of $\pm 500$ m Sofiev et al. (2012). For each fire, we calculate the emission uncertainty

$$\Delta E_{\text{lag}}^{p/m} = \left| \frac{E_{\text{lag}}^{p/m} - E_{\text{lag}}}{E_{\text{lag}}} \right| \tag{10}$$

using plume heights $z_{\text{lag}}^p$ and $z_{\text{lag}}^m$, respectively (see Fig. 5(b)). Analogously, the uncertainties $\Delta E_{\text{c}}^p$ and $\Delta E_{\text{c}}^m$ for $E_{\text{c}}$ are computed. The uncertainties change from fire to fire and can be found in the data (Goudar et al., 2023). Table 2 shows the value of the largest uncertainty per region. For the Siberian region, the maximum uncertainties are small, indicating little vertical variation in the velocity. For the US and Australian regions, the corresponding uncertainties are much larger and the uncertainties corresponding to the constant emission height are a factor 2-3 times higher than the Lagrangian plume height uncertainties. This hints at a more variable wind field for these regions. Overall, we estimate that this APE error term is the largest error contribution with an error $< 100$ % for each fire.

Finally, we consider systematic errors in wind velocity that are constant in the plume domain. The error propagates one-to-one into the error of the flux estimate. Uncertainties of the ERA5 wind fields in the tropospheric boundary layer are not reported. Gualtieri (2022) derived surface near wind errors of $1.76 \text{ ms}^{-1}$ (root mean square error) for ERA5 data. A typical wind speed at the plume height is 3-11 $\text{ms}^{-1}$, and although at the plume height, the wind speed error of $1.76 \text{ ms}^{-1}$ might be smaller, we consider this error as a significant error contribution. However, we refrain from quantifying this error due to the lack of reliable knowledge.

### 3.2.4 Emission uncertainty in APE

We verify our uncertainty estimates by evaluating WRF simulations of a CO plume using APE. The WRF simulation was performed using real atmospheric forcing at 1 km resolution for a fire with the highest FRP (USA, 12 September 2020; see Sect. 3.1). The details of the WRF simulation can be found in Appendix A. Three plumes at three different UTC times shown in Figs. 9(a)-(c) were selected, and emissions were estimated by our algorithm. It should be noted that the averaging kernels were not used to degrade to TROPOMI data, and only the enhancements were simulated in the model, thus the background is set to

**Table 3.** Comparison of actual emissions to the emissions computed at plume height $z_{\text{lag}}$ for the three selected plumes shown in Fig. 9. The uncertainty in the table is computed as 100*(Actual - Computed)/Actual.

| Time in UTC (H:M) | Actual (kgs$^{-1}$) | 1 km grid (kgs$^{-1}$); Uncertainty | TROPOMI grid (kgs$^{-1}$); Uncertainty |
|---|---|---|---|
| 17:00 | 28.45 | 20.26;    28.8 % | 20.67;    27.3 % |
| 18:00 | 56.84 | 34.92;    38.5 % | 34.52;    37.5 % |
| 19:00 | 97.86 | 99.15;    -1.5 % | 99.36;    -1.53 % |

zero by the simulation. The plume height ($z_{\text{lag}}$) was calculated as the maximum height where the concentration became zero, and the fire sources were the same as the sources used in the WRF simulation. The velocity used in both Lagrangian simulations

and emission estimations was inferred from the WRF velocity data. The emissions for these plumes were estimated by APE assuming a constant emission in time and are presented in Table 3. Here, the actual emission is the mean of the total known CO emission from all fire sources with time. The averaging interval is defined as the time the particles take to reach the final transect. Additionally, we degraded the simulation grid to the TROPOMI grid shown in Figs. 9(e)-(g).

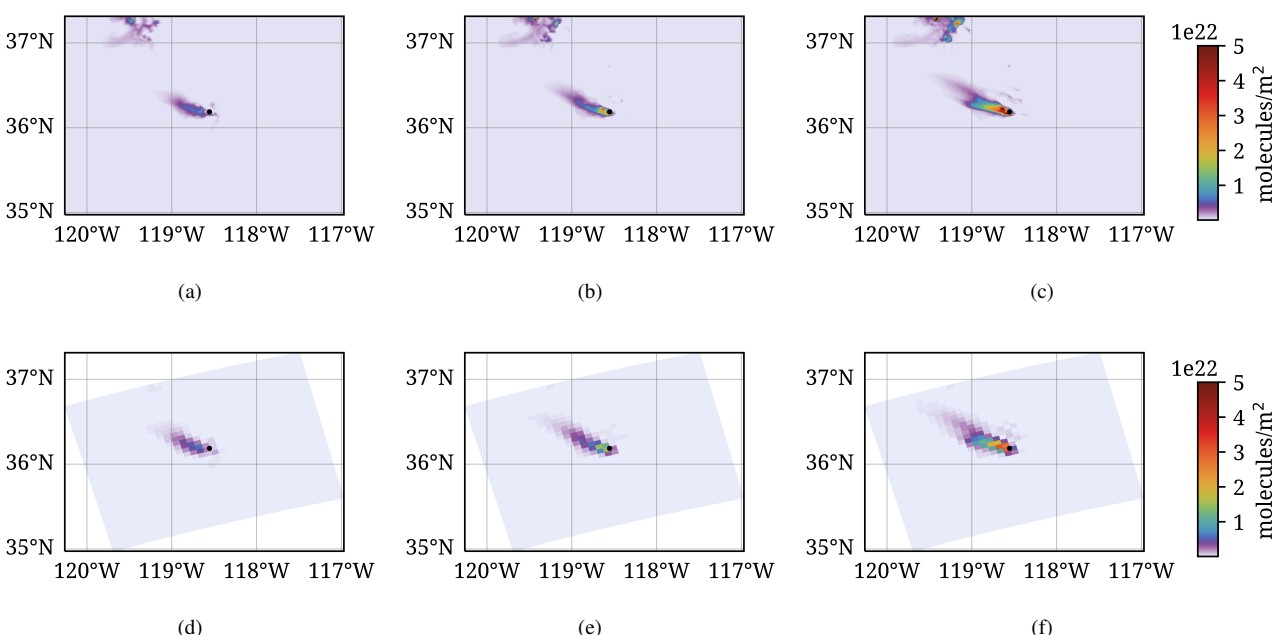

**Figure 9.** Three selected plumes at three different UTC times (a, d) at 17:00, (b, e) at 18:00 and (c, f) at 19:00. Top (a-c) represent plumes at 1 km grid resolution and bottom (d-f) represent TROPOMI grid resolution.

The uncertainties of the APE emission estimate range between -1.5 % and 38.5 % (Table 3). In all three plumes, the velocity

and plume height used by APE is appropriate, however, the emissions computed by our algorithm differed from the actual

emissions. This is attributed to the error in the cross-sectional flux method due to the assumption of constant emissions, which might not be the case for a fire. It should be noted that this uncertainty is for one particular case, and it can vary depending on the case. For the three selected cases, the CFM method leads to an error of 28.8, 38.5 and 1.5 % which is in the range of the derived standard error. The difference in emissions between high resolution (1 km grid) and low resolution (TROPOMI grid) was found to be less than 2 %. If the velocity is accurate, then it can be concluded that having higher-resolution data does not have much effect on the cross-sectional flux method. Overall, this analysis suggests that the assumption of a constant emission is the major error source next to errors in the wind field and uncertainties in the injection height.

## 4    Conclusions and recommendations

An automated plume detection and emission estimation scheme for CO flux inversion for single point fires was developed by integrating four freely available data sources: the VIIRS active fire dataset, the TROPOMI CO dataset, the injection height from GFAS and ERA5 meteorological data. The automated plume detection and emission estimation algorithm (APE v1.1) was optimised for one region, and its performance is verified for three months of data for two other regions, Australia and Siberia. For all regions and for all the fire sources identified by VIIRS, 16 % (882 cases) of the data correspond to clear sky TROPOMI CO data with a plume signature. Out of those 882 plumes, 309 plumes were too short and about 195 had multiple sources of fire in them. Internal quality filtering of APE reduced the number of estimated fires to 226 cases which is 26 % of the 882 cases. Finally, the visual filter on APE's output of 226 cases showed a true-positive confidence level of 97.7 % (221 cases). One key element of automated APE detection of fire plumes in the TROPOMI CO dataset is prior knowledge of potential fire locations coming from the VIIRS active fire data product from Suomi NPP. It highlights the potential to fly the Suomi NPP and SP5 satellites in a loose formation with a temporal separation of 3.5 minutes.

To estimate CO fire emissions, we employed the CFM. Here, we considered three different assumptions on plume heights, first a constant plume height at 100 m altitude, second a constant plume height at the GFAS injection height, and third a varying plume height using a Lagrangian model. The varying plume height approach best reflects the characteristics of fire. If a fire is at its peak, strong convection leads to an upsurge of air, and at the same time, it is transported downwind from the fire source. Note that we assume that the ERA5 velocity fields incorporate this heating effect to some extent, as it assimilates the surface temperature observed by satellites. In our simulations, the plume height varied by more than 500 m in a downwind direction for 30 out of 221 cases, and all 30 cases were in the United States and Australia. The variation in plume height was found to be minimal in Siberia.

The assumption of plume height at 100 m led to unreliable emission estimates and was discarded. The difference in estimation emission for the constant injection height of the GFAS and the varying plume height was observed to be less than 4 % for the Siberian region. We observed larger differences for the US and Australia, where the maximum uncertainty using the varying plume height is half that using a constant plume height. Based on these findings, we decided to use the Lagrangian model for plume height as the processing baseline for APE.

Overall, we estimate the uncertainty of our product with a standard error of $< 20$ %, which mainly accounts for errors due to spatial under-sampling of the CO field by TROPOMI and the assumption of constant emission for the time frame relevant to plume formation. The TROPOMI CO data are of high quality with respect to precision and bias. Based on TCCON, the TROPOMI CO data do not provide any significant contribution to the emission estimate of APE. Additionally, we analysed emission errors due to the uncertainty of injection height from GFAS. Depending on the meteorological situation in the different regions, errors are $< 100$ %. The random error in the meteorological data (wind velocities) was described using the ERA5 ensemble data and was found to be less than 20 % for 97 % of the cases. Systematic errors due to the wind for every fire case were also considered important, however, they cannot be specified, as the ERA5 data product does not provide an estimate of systematic wind errors.

Finally, the presented APE algorithm is appropriate for estimating CO emissions from single isolated fires from TROPOMI and VIIRS data using a fully automated algorithm. It is considered a baseline for future APE upgrades to optimise automated emission estimates of CO point sources. As a next step, we consider (1) the processing of the entire CO TROPOMI data set, (2) expanding emission estimations for multiple fire sources, (3) developing an improved inversion scheme, which can be done by developing algorithms that map the simulated tracer particles from Lagrangian simulations to the TROPOMI CO concentrations to compute emissions and (4) comparing the emissions predicted by APE to available emission databases.

## Appendix A: The WRF Model description

The WRF model configured in a two-domain configuration is applied in the tracer mode to simulate the transport and dispersion of CO emitted by a wildfire in the US. The outer and inner domains are run at a horizontal grid spacing of $5 \times 5$ km$^2$ and $1 \times 1$ km$^2$, respectively. The model domains are centred at $36.16225^0$ N, $119.1528^0$ E and have 43 vertical levels that extend from the surface to a model top of 50 hPa. The outer domain has $200 \times 200$ grid points, while the inner domain has $400 \times 350$ grid points in the west-east and north-south directions. The meteorological initial and boundary conditions for the outer domain are based on the Global Forecast System (GFS) forecasts available every 3 hours at a horizontal grid spacing of $0.25^0 \times 0.25^0$. The static geographical fields and the GFS output are mapped onto the WRF domains using the WRF pre-processing system (WPS). The physical parameterisations follow Kumar et al. (2021), except for the cumulus parameterisation that is turned off in the inner domain.

Biomass burning emissions are obtained from the NCAR Fire Inventory (FINN; Wiedinmyer et al., 2011) version 2.5 and are distributed vertically online using a plume rise parameterisation developed by Freitas et al. (2007). This parameterisation selects fire properties appropriate for the land use in every grid box containing fire emissions and simulates the plume rise explicitly using the environmental conditions simulated by WRF. Since we are using the model in the tracer mode, the chemical evolution of the plume is not simulated. To describe the loss of CO in the model, we allow the CO fire emissions to decay with an e-folding lifetime of 30 days. No other source (anthropogenic emissions, biogenic emissions or photo-chemical production from hydrocarbons) is included in the simulation. The model run started on 12 September 2020 at 12 UTC and stopped on 13

September 2020 at 00 UTC. We used a time step of 20 s for the outer domain and 4 s for the inner domain. The model output
is saved every minute and used for further analysis.

## Appendix B: Algorithm and Simulation details

---
**Algorithm 1** APE algorithm: Pseudo-code

---
**Require:** region and time
  **for** region and time **do**
    Find Fire sources from VIIRS data (Sect. 2.1.1)
    **for** Each fire source **do**
      Extract TROPOMI CO data granule (Sect. 2.1.2)
      **if** Data is good **then**
        Detect plume by plume detection algorithm (Sect. 2.2)
        **if** Plume is detected **then**
          Estimate emission (Sect. 2.3)
        **end if**
      **end if**
    **end for**
  **end for**

---

**Table B1.** Considered region and time. The region is rectangular and is constructed based on the origin and width and height. The origin is always the south-western point of the region.

| Label | Region Origin [(lon, lat)] | Region Size (Width, Height) | Time | L2 product version | VIIRS, ERA5 and GFAS Data access |
|---|---|---|---|---|---|
| US | $140^0$W, $20^0$N | $80^0$, $45^0$ | Sept 2020 | 1.03.02 | 10 Oct 2020 |
| AU | $70^0$E, $53^0$S | $55^0$, $27^0$ | Oct 2019 | 1.03.02 | 10 Oct 2020 |
| Sib | $113^0$E, $44^0$N | $41^0$, $34^0$ | June 2021 | 1.04.00 | 5 Feb 2023 |
| Sib | $113^0$E, $44^0$N | $41^0$, $34^0$ | July 2021 | 2.02.00 | 5 Feb 2023 |

*Code availability.* APE v1.1 code is archived on Zenodo (DOI:10.5281/zenodo.7740542).

*Data availability.* The TROPOMI CO dataset of this study is available for download at ftp://ftp.sron.nl/open-access-data-2/TROPOMI/tropomi/co/
(last access: 5 Feb October 2023). The IS4FIRES injection height and the 3-d velocities at 127 model levels were obtained from the

**Table B2.** Filtering from fire clusters to good CO data. The total column is the same as the fire clusters in Table 1.

| Region | Grid size | Quality | Multiple Clusters | Good Data | Total |
|---|---|---|---|---|---|
| US | 442 | 373 | 53 | 213 | 1081 |
| AU | 1020 | 512 | 87 | 385 | 2013 |
| Sib Jun | 37 | 249 | | 130 | 416 |
| Sib Jul | 34 | 1419 | | 599 | 2052 |
| All Regions | 1533 | 2553 | 140 | 1327 | 5562 |

**Table B3.** Filtering from good data to plume detection Table 1. The total should represent the good CO data available.

| Region | No enhancements | Short plumes | Other clusters | Detected plumes | Total |
|---|---|---|---|---|---|
| US | 42 | 41 | 51 | 79 | 213 |
| AU | 57 | 62 | 94 | 172 | 385 |
| Sib Jun | 22 | 25 | 12 | 71 | 130 |
| Sib Jul | 324 | 181 | 38 | 56 | 599 |
| All Regions | 445 | 309 | 195 | 378 | 1327 |

**Table B4.** Filtering from plume detection to emission estimation Table 1. The total should represent the plume detection cases.

| Region | No Injection height | Background Subtraction | Plume alignment | Velocity $< 2\,\mathrm{ms}^{-1}$ | Emission estimation | Total |
|---|---|---|---|---|---|---|
| US | 14 | 14 | 11 | 3 | 37 | 79 |
| AU | 20 | 4 | 13 | 6 | 129 | 172 |
| Sib Jun | 15 | 2 | 15 | 4 | 35 | 71 |
| Sib Jul | 8 | 9 | 12 | 2 | 25 | 56 |
| All Regions | 57 | 29 | 51 | 15 | 226 | 378 |

Global Fire Assimilation System (GFAS) database and the European Center for Medium range Weather Forecasts (ECMWF) Reanalysis v5 (ERA5), respectively on 10 October 2020. The Visible Infrared Imaging Radiometer Suite 375m thermal anomalies/active fire product was also accessed on October 2020 (https://firms.modaps.eosdis.nasa.gov/active_fire/). The processed data is available as DOI: 10.5281/zenodo.7738734.

*Author contributions.* MG developed the code and performed the analysis with few inputs from TB and JL. JA master's thesis served as a feasibility study for this work. RK performed WRF simulations and gave inputs on its data analysis. All co-authors commented and improved the paper with a special mention to JL and MG.

*Competing interests.* The authors declare that they have no conflict of interest.

*Acknowledgements.* The National Center for Atmospheric Research is sponsored by the National Science Foundation. The authors thank
Andrew Barr for proofreading the article.

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
