# Peer review of "Plume detection and estimate emissions for biomass burning plumes from TROPOMI Carbon monoxide observations using APE v1.1"

_EGUsphere, 2022_

## Author Comment (AC1)

Dear Anonymous Referee #1,

We would like to thank the referee for his/her constructive comments and recommendations on our manuscript. We have done few major revisions on the manuscript. Our replies to the comments are given below. The original comments of the referee are numbered, given in black and the answers are given in blue. The adjustments in the revised manuscript are specified in detail below where the page and line numbers refer to the revised manuscript.

**Software changes APE v1.0 – APE v1.1:**

(a) We have shortened plume length from 40km to 25km. (Filter PD-1 in manuscript)

    a. Lead to increase in detected plumes from APE V1

(b) Some minor bugs fixes and re-wrote few section of code. No effect on results.

**Comments**

1. "One main comment is that the derived emissions are not compared to any other CO emissions estimates, so it is difficult to judge how good these are. I understand that this could perhaps be the subject of a separate paper, but this is not mentioned by the authors. I would suggest that the authors prepare supplementary material with the estimated CO emissions e.g., in the form of a spreadsheet so that the values could be compared with other emission values by others."

   Changed:

   a. Added to the manuscript in the results section, see Lines 339-341
   b. We have added the suggested content to the manuscript as a reference doi:10.5281/zenodo.7728874. Additionally, the referee can view the plumes mentioned in the paper in https://emhelium.users.earthengine.app/view/firespaper

2. "p3: You list 4 methods but discuss only three. Please discuss the missing one: IME."
   a. Changed: We have added details on IME to the revised manuscript (Line 60).

3. the choice of r_max=4km is puzzling. Will it not automatically discard the megafires from the analysis? Maybe this should be emphasized in the text?
   a. Changed: The focus of the current version of APE is to quantify emissions of isolated hot-spot fires. This aspect is stressed now at several places in the manuscript (Line 74, 175-176, and 406) So, if mega fires mean spatially extended fires, these are not addressed with the presented version of the algorithm. However, strongly localized fires are addressed.

4. Figure2. I don't understand this figure. What is the message? Is this supposed to be good?
   a. Changed: We removed the figure. It causes more confusion than expected.

5. Many fire-counts are not considered by DBSCAN. Why? Because there are less fire counts than n_min?
   a. No Changes: Yes, there are fewer fire counts than n_min. Results of this work in Table 1 also shows that even more than 10 fire counts do not lead to an atmospheric signal that can be detected in TROPOMI.

6. The selection of fires does not consider any criterion on the fire intensity (FRP). Why (not)?

a. No Changes: Yes. The assumption is FRPs might be low for a fire which is at its end. And the old fires might have strong plume signal. We need to investigate this further; we have planned future work on this subject.

7. Also noticeable are the fire counts over sea. To what these pixels correspond?
   a. Not changed: Figure 2 is already removed. The reason for counts of see is probably due to refineries or some burning events, whereas we cannot exclude false flags.

8. Figure 4 shows a relatively isolated CO plume but how is the plume detection working for the other plumes close to each other?

   No changes: We think this is a misunderstanding. Plumes can and are detected even if they are next to each other. This is not a limitation of the data yield. However, if plumes are too close data will be rejected due to multiple fire sources because of difficulties to determine the atmospheric background (Filter PD-2 in the manuscript).

9. p7, l 142: What is a 'connected region'. What is the CO VCD criterion related to this?
   a. Changed: Added an explanation in the manuscript. Lines 155-158.
   b. CO VCD is not related to this as the detected plume is only used to draw plume lines.

10. Section 2: section 2.3.1 :- Is the re-centering needed? Or is to facilitate the Gaussian fit? Please clarify
   a. Changed: Yes, it is to facilitate gaussian fit and has been added to the manuscript. Lines 206-207.

11. p11 l 219: what is the name of the model used for the simulations? Is it defined somewhere?
   a. Changed: The model is a Euler forward model and an appropriate reference is added. Lines 239-240.

12. The authors attempt to account for wind variability in the horizontal and vertical dimensions. However, there is an additional flux term due to the partial derivative of the wind which is not accounted for (see the divergence method of Beirle et al., Sc. Adv, 2019). Can you quantify this?

   No Change: Beirle et al.'s method aims to quantify multiple sources that have fixed geo-location and average it temporally. Our focus is on preselecting isolated plumes of single overpasses so that multiple sources are rejected as well as possible prior to the flux inversion.

13. Section 3: P12: Going from 622 to 196 plumes is in a way disappointing. Does that mean that only ~1/3 of the fires made a meaningful CO signal in the TROPOMI data? Please elaborate.

   Changed: Details on how plumes get filtered are now added to the manuscript. Sec 3, P12, L283-287. This happens due to three reasons: Some data will have no meaningful signal because of the TROPOMI detection limit, short plumes that cannot be interpreted with CFM, and plumes have multiple fire clusters in them. See Table B3 in appendix for details.

14. The discussion on errors should be expanded. The error characterization based on standard error (Eq.5) does not account for any systematic error and mixes random errors and real CO flux variability, so it is not a very good metric. I would propose including a table summarizing all error sources and estimating them.
   a. Partly changed: Our philosophy is to use the standard error to characterize errors that this quantity is sensitive. In addition, we discuss other error sources like TROPOMI CO biases, injection height, and wind speed. The section is revised and hopefully also improved.

b.  Yes, we agree. We changed the Uncertainty estimation section. See Sec 3.2.

15. Section 3.1.1: Generally, zlag seems higher than zc which is in contradiction with Fig 7b. It is confusing. Perhaps it is due to an unfortunate choice in the illustration?

Changed: Yes, an unfortunate choice in the illustration. Added a new Fig. Now it is Fig 5b in revised manuscript.

16. P13, l273: the author states: "a relation between plume height rise and these two variables can be expected as higher FRP means higher temperature which heats up the air, leading rise of the warm air." However, this process of self-heating is likely not accounted for in the Lagrangian modeling. In fact, the approach presented here is in fact limited to a certain range of fires not too low (because of the limit of detection of the satellites) and not too big (because self-heating and other non-linear processes are not well represented). Therefore, Fig 9b is misleading. The differences are very small, but it does not mean Ec is good because the Elag is not representing all the physics.

a.  Partly changed: Yes, that is true. And we do not include heating in Lagrangian simulations, but we assume that the ERA5 velocity fields incorporate this heating effect as ERA5 assimilates the skin surface temperatures from satellites. And this has been added to manuscript (Lines 248-249)

b.  Including heating in Lagrangian simulations is considered to be future work.

17. -P4, l91: 'Mostly, an emission plume created by a burning'->'Essentially, a plume emitted by a fire'. The sentence states that a fire in a single VIIRS pixel cannot be detected by TROPOMI. Why not? On what is based such statement?

No Changes: What we meant here is that it is difficult for a single pixel (0.14 sq km) to create a plume spanning a few TROPOMI pixels. The limitation is the detection limit of TROPOMI because of the pixel size and precision of the measurement. We observed 5562 fire clusters (at least 10 fire counts) and good TROPOMI data is only observed for 1327 fire clusters because of this limitation.

18. -p18, l370: 'reliable' is subjective. You don't have any way to assess whether it is more reliable or not.

Changed: We agree on this. This text has been rewritten.

19.  P18, l384-387: What about overlapping plumes from different fires? Isn't there a way to improve on this?

No Changes: We believe that something like Beirle et al. method is possible. Our approach was first to get something in place that works in an automatic way for the 'simpler' cases and to improve the approach later if possible.  So, we foresee this for the next iteration of the APE algorithm.

20. TYPOS/text Suggestions

a.  -acronyms are sometimes defined multiple times. Please define acronyms only once.

i.  Changed.

b.  -Both acronyms 'Tropomi' and 'TROPOMI' are used in the text. Please use one or the other throughout the text.

i.  Changed.

c.   -several suplots /maps have no units. Please define the units for all figures.

i. Changed.
d.   -several figures or subplots would be better placed in the supplementary material: Figs
    7ce, Fig 8.
    i. Changed: 7c-e are removed. Fig 8 is still left in the manuscript. Now Fig 6.
e.   -P2, l27: 'CO in atmosphere and Shi et al.' -> 'CO in the atmosphere. Shi et al.'
    i. Changed.
f.   -P2, l30: 'has been on increase' -> 'has been increasing'
    i. Changed.
g.   -P2, l37: 'between two measurements' ->'between the two measurements'
    i. Changed
h.   -P2, l50: refer to the use of VIIRS for methane cloud masking does not help the clarity of
    the text.
    i. Changed: Removed from the manuscript.
i.   -P3, l74: 'deliberated'-> 'discussed'
    i. Changed.
j.   -p4, l107: 'constrained' -> 'restricted'
    i. Changed: Paragraph has been reworded.
k.   -p6, l129: l129: Gaussian filter : is this a 2D convolution?
    i. Changed: More details have been added to the manuscript.  Lines 142-143.
l.   -Fig5d is not appearing in the manuscript.
    i. Changed.
m.   -p18, l362: doesn't -> does not
    i. Changed. During the rewording of the paragraph.

---

## Author Comment (AC2)

Dear Anonymous Referee #2,

We would like to thank the referee for his/her constructive comments on our manuscript. We have done few major revisions on the manuscript. Our replies to the comments are given below. The original comments of the referee are numbered, given in black and the answers are given in blue. The adjustments in the revised manuscript are specified in detail below with line numbers referring to the revised manuscript.

**Software changes APE v1.0 – APE v1.1:**

   (a) We have shortened plume length from 40km to 25km. (Filter PD-1 in manuscript)
       a. Lead to increase in detected plumes from APE V1
   (b) Some minor bugs fixes and re-wrote few sections of code. No effect on results.

**Major comments:**

1. unclear how "fully automated" is this method. Numerous thresholds are applied but their values are not justified. Also: how universal are the numerous threshold values required by the method? How reliable are the results when applied to other locations and times?

   Thresholds were optimized for US and we observe that it generalizes well. So method can be applied without any visual inspection of intermediate data. We have applied the algorithm to different regions and times. Australia (Oct 2019) and Siberia (June-July 2021) analysis in the manuscript. Major changes have been made to the manuscript.
   Changed:
        i. The reasons for the choice of thresholds have been given (Sec. 2)
        ii. Added region encapsulating Siberia for months of June-July 2021 (mainly Sec 3)

2. unclear what percent of all VIIRS fires are detected and their emissions successfully quantified with this method. The quoted 97.9 % success rate seems too high (considering that many plumes seem to be rejected due to different reasons) and is not properly justified.

   This is the wrong perspective in our opinion. We do not want to claim that we can detect all fires detected by VIIRS. What the paper quantifies are the fires that can be observed in TROPOMI CO data with respect to pre-selected VIIRS data.
   Changed:
   • It is important to show the false positive detection by the algorithm and the 97.0 % gives the confidence level of the detected data. This is now phrased properly (Line 297)
   • The reasons for rejection have been explained in detail in the revised manuscript (Lines 276-297)

3. unclear how valid the plume height values and the emission values quoted in the manuscript are, since no attempt was made to validate those with respect to in situ measurements
   • No Changes: Reference to the GFAS data including its uncertainty is given in the manuscript. Sec 2.3.3 and lines 224-228

4. Specific issues: details needed for analysis replication are missing, e.g., filter size, filter formulation.
   • Changed: Filter criteria are explicitly mentioned in the revised manuscript (DD-1, DP-2, PD-1, PD-2 , EE-1 to -4, see Sec. 2)

5. Specific Issues: tools are not described and, thus, become black boxes to the reader.
   • No changes: References to all numerical tools used are given in the manuscript. These are available to the reader. The manuscript describes the main functionality of the tools. We think this is common practice and an appropriate approach. We do not know how else an algorithm description can be provided as the use of libraries is a common practice.

6. 16: "The emissions were severely under-predicted".
   a) With respect to what?
   b) Were there any in situ measurements used to validate the emissions calculated here?
       a) Changed: We have changed the way the results are discussed. (Lines 330-339)
       b) Not changed: To our best knowledge, there are no independent validation measurements (in-situ) available. On one hand, this represents the limitation, on the other hand, demonstrates the novelty of the data product.

7. 22: Please clarify "idealized cases"
   • Changed: Removed during the re-write of the manuscript.

8. 25-26: please provide a reference for "it is a weak greenhouse gas"
   • Changed: "Weak" is replaced by 'indirect' to represent what was intended, reference (Spivakovsky et al., 2000) is given. See line 26.

9. 91-92:
   (a) Could missing fire counts (due to missing VIIRS pixels because of, for example, clouds/smoke) result in fires and, thus, in plumes not being identified by the automated plume detector?
   (b) Some CO plumes may only be detectable downwind from the fire, if clouds/smoke mask the fire, which is not uncommon.
      a) No changes: Yes, that is a possibility.
      b) Yes, some plumes are detected in downwind from the fire. This is one of the reasons why we detect plumes within certain pixels from the fire source pixel (see Lines 168-169).

10. 98: Please correct to "The minimum number […] has been set to $n_{min}$=10" Also, why 10?
    - Changed: Thresholds like $n_{min}$ are tuned for US fires and verified for Australia and Siberia. (Sec.2.1.2, line 121 and sec 3)

11. 105: a granule would be much larger than 41 x 41 pixels; please consider using "subset" instead of "granule" here as well as in lines 107, 108, 109, 110, 111, 114, 128, and Fig. 3 caption. (A granule would have whatever size is covered by a whole TROPOMI file.)
    - No changes: We believe 'granular data' is data that is in pieces, as small as possible to be more defined and detailed. So, we think it is the right wording.

12. 110: how were the 80% and 85% thresholds selected? Also, please explain the meaning of QA>0.5
    - Changed: The threshold was selected to disentangle the plume signature from the background Added to the manuscript. See lines 116-120.

13. Fig. 2: "fire-counts that were not clustered" was it because there were less than 10 fire counts within a 4 km radius? Please explain
    - Changed: See item 10.

14. Also: why 41x41 pixels? Why 7x7 pixels?
    - Changed: A motivation is added to the manuscript. If a large number of pixels are bad in 41x41 pixels and if they are around the fire source, then no plume will be detected. That is the reason for 7x7 pixels. (liness 111-113)

15. 116: please explain "gold standard data"
    - Changed: Removed and reworded (lines 126-128)

16. 121: "Thus, the watershed algorithm segments the regions into valleys and mountains (CO enhancements) based on a given marker" Valleys suggest low CO regions and mountains high CO regions, i.e., plumes. However, line 125 seems to say that what the algorithm does is to look for low/high boundary zones, i.e., zones of maximum slope change. Please clarify.
    a. Partly changed: The section has been reworded for clarity (Sec 2.2)
    b. The algorithm requires two inputs, gradient map and marker image containing seeds of high and low CO, to segment (see changes in Sec 2.2)

17. How does this method perform compared to simply calculating the background value in a TROPOMI scene and then selecting clusters of pixels above that value?
    - No change: We think the referee underestimates the heterogeneity of the CO background field. Due to the moderately long lifetime of CO, the background for a location is always changing and it can contain emission features from other sources and long range transports. A simple subtration of background does not work.

18. Also, it looks like only one of several plumes in this 41x41 TROPOMI subset is detected, even though several fires are shown in the same 41x41 subset in Fig. 3; please clarify what happened to the other plumes in this subset, including the largest of them all, in both size and CO value: did the algorithm identify all of them or just one of them?
    - Changed: We understand from this and from few follow-up comments below that our explanation is not clear. Thus, we have done necessary additions to the manuscript. Furthermore, a pseudo-code of APE algorithm (Appendix B) is added to clarifies the idea further. Also have explicitly mentioned this in lines 124-125.
    - Each extracted 41x41 granule corresponds only to one fire source.
    - All plumes were detected since they were assigned to different data granules.

19. Fig 4c-4d: plumes detected are much shorter than the actual plumes.
    - Partly changed: Quantification in downwind direction is difficult due to lower enhancements in downwind direction and heterogeneity of the CO background. Therefore, we only consider strong enhancements in the plume. See lines 170-172.

20. 128-162: Please clarify if this example illustrates the process followed to either 1) identify a single plume in the 41x41 TROPOMI subset; the process is then repeated for each of the remaining plumes in the subset or 2) all plumes in the 41x41 TROPOMI subset at once. If 1) is true: please clarify text. If 2) is true: most plumes are missed, please discuss.
    - Changed: See item 18 (above).

21. 129: "First, high frequency components of the CO-image are reduced by a Gaussian filter" Please explain, is that to remove noise? What is the size of the filter, is the size constant for all plumes, how was it selected
    - Changed: This is used to reduce the noise. We added an explanation in manuscript. Lines 141-142.
    - Standard deviation of Gaussian filter $\sigma = 0.5$ (in pixels) (added to manuscript) and yes, the size is constant for all plumes. This was an emperical choice.
22. 130: "the elevation map Ielev is computed using a Sobel operator" Describe with an equation what the Sobel filter does. Also, "elevation" seems incorrect here, since the Sobel filter would highlight zones of maximum change in slope in the input. Consider changing to Isobel or similar.
    - Changed : Referred as I_grad in the manuscript and have included the equations to Sobel opeators. Line 144-147.
23. 134: stating that Imark is initialized with zeroes would suffice, no need for an equation.
    - Changed : done.
24. 136-140: for clarity and simplicity, consider rewording to "[…] clear CO enhancement. Pixel Imark(i,j) is considered CO enhanced (i.e., Imark(i,j)=2) if Is(i,j) is either above the median of Is or above the mean of the 15x15 pixels centered at Is(i,j). Otherwise, Imark(i,j)=1. For our example in Figure 4 [...]" (no equations needed). Also: why 15x15 pixels? Is this size fixed, or does it change from plume to plume?
    - Changed: We have changed the paragraph and reworked on it. See from lines 151.
    - 15 x15 is again empirically chosen to account for background variability. This has been added to manuscript (line 152-155).
    - This remains the same for all plumes.
25. 140-141: the meaning of the last sentence in the paragraph is unclear. It looks like the result of the step that was just described (where Imark is populated with either ones or twos) is illustrated by panel 4d, not 4c. What is panel 4c? How is it relevant? Please comment on the plumes present in 4a and 4b but absent in 4d; one of the absent plumes was the largest of them all, in both size and CO values.
    - Changed: Rewritten for more clarity in the manuscript. Lines from 155.
    - Each plume corresponds to a fire source and is detected separately. See item number 18 for further clarity.
26. 142: how does this new tool work?
    - Changed: 'label' algorithm identifies all connected pixels with the same value and creates a region. A description has been added to the manuscript. See lines 156-158.
27. How is this plume detection algorithm better than a simpler approach, such as identifying groups of CO pixels with values above that of the background? It looks like the latter would have sufficed to identify all the plumes in this 41x41 TROPOMI subset, while this plume detection algorithm (at least according to Fig. 4) missed most of them.
    - Changed: See item 17 and 18.
28. 169. (Here and elsewhere in the manuscript) wind velocity from ERA5 data is expressed in the manuscript as "u". Usually (and that includes the ERA5 dataset) u represents the E-W component of wind; v would represent the N-S component. Is u in the manuscript really the E-W component of the wind? Shouldn't the wind velocity be calculated according to the plume's direction? Please clarify
    - Changed: Added new symbol in the manuscript (see equation 2) to get rid of confusion.
29. 175 Please clarify "The plume line results from a second order curve fit through the pixelcenters of the identified pixels"
    - Changed: The vague wording of "identified pixels" to "detected plume" in the manuscript.
    - Plume line is the solid black line in figure 4a, along the plume. See Line 196.
30. 176 Why 2.5 km?
    - Changed: To reduce interpolation errors. Added to the manuscript. Lines 197-198.
31. 177 Why 500 m?
    - Changed: Oversampled to get smoother CO distribution. Added to the manuscript. Lines 199-200.
32. Fig. 5 caption: c): why missing value at 0 km from the source?
    - No Change : The pixel at the fire source has a qa value < 0.5. Usually, we do find the pixels near the fire have bad quality due to smoke.
    - Changed: Figure has been changed. Figure 4 in revised manuscript.
33. 182. Please explain what are the terms H0, H1, and A0
    - Changed: H0, H1 refer to background and A0 to intensity. Added in the manuscript, Line 213.
34. 203. Fig. 6 shows two distinct plumes approximately 100 km long each, resulting from two fires 50 km apart. According to the text, both plumes were rejected by this algorithm because they were too close to each other. How close is too close? What's the minimum plume size detectable with this method? These and other limitations of the method presented in this manuscript should be discussed both in the abstract and in the conclusions sections.

- Not Changed: 'Close' cannot be quantified in distance but in terms of a too heterogeneous background. This is done in the manuscript. Section 2.3.2.

35. What are the dashed lines in Fig. 6?
    - Changed: Figure removed. They represented transects.

36. 207. Please explain briefly why is the uncertainty in injection height about 500 m.
    - Partly Changed: We quote Sofiev et al. (2012), who showed how the IS4FIRES injection height deviations from MISR Plume Height Project (MPHP). Has been added to the manuscript. Lines 228-230.

37. 224. Why 6 hours?
    - Changed: This is chosen based on diurnal cycles. Added explanation to the manuscript. Lines 248-249.

38. Fig. 7 b) values at distance=0, 2.5, and 5 km from the source (i.e., at the fire source and close to it) are missing; please explain. The text states elsewhere (e.g., l. 366) that plumes go higher away from the source but the opposite behavior is shown here.
    - Changed: See item 32.
    - Figure 7 has changed in the manuscript as it is an unfortunate choice in the illustration. Now, Figure 5 in revised manuscript.

39. 234-245. Please quantify what proportion of plumes are rejected due to: lack of GFAS injection height, disagreement between Lagrangian particles flow direction and actual plume direction, wind velocity below 2 m/s.
    - Changed: Details added to the manuscript. See section 3 and Appendix B shows all the rejected cases in detail.

40. 260: "To conclude, presented automated algorithm can successfully detect plumes and compute emissions for ≈ 97.9% of the cases." It looks like the percentage of plumes detected is much lower that that. How was this figure calculated? A few sentences earlier the text says "the plume detection algorithm [...] identified 196 plumes among 622 cases" and lists numerous cases which were not successfully processed due to a number of reasons. The detection rate quoted does not seem feasible, unless relevant qualifiers are missing.
    - Major changes: Section 3 has been rewritten. Now we describe how data are filtered in detail.
    - We agree with referee and we now speak of "confidence level" of the APE algorithm.

41. 264 How is it decided what is the number of transects along the downwind direction to beconsidered? Does this number change from plume to plume, or is it universal?
    - No changes: Number of transects depend upon the plume size and does change from plume to plume.

42. 3.1.1. As expected, emissions calculated using plume height Zlag and Zc differ when the heights themselves differ. Unclear if results vary from Australia to USA; thus, please consider using the same symbol/color for data points from both locations. Unclear if all 4 panels are relevant; some seem redundant. Consider showing one panel with height difference (between zlag and zc) versus emissions difference and another panel with height difference (between zlag and fix z=100 m) versus emissions difference.
    - Changed: We have made new figures and added new data from Siberia. Additionally, text in manuscript has been updated. See Sec 3 and Sec. 3.1.

43. 277-287. Unclear where the discussion is going until the last sentence "although the overall effect of the Lagrangian estimate of the plume height on the emission estimate is minor, we could identify several cases where the emissions estimate become more reliable." Consider starting the paragraph with this sentence and add a very brief description of relevant data.
    - Changed: We have re-organized the manuscript and this has been included. From line 319.

44. 295. 10% change in emissions seems to be much smaller than some of the emission uncertainties discussed later on (e.g., l. 344-345). Also, a 10% variation in emissions was qualified as "minor" elsewhere (l. 19). Please discuss.
    - Changed: We now motivate that it is difficult to scale 100m winds to compute emissions at plume heights (injection and varying plume height). This part of manuscript has been re-written. Paragraph starting at line 332.

45. 325. "It should be noted that this uncertainty has been reduced to 3.4ppb in the newer versions of L2 product". How much is that in percent value?
    - No Change: Depends on background concentration and means typically 3-10 %. We do not want to use relative errors in the manuscript.

46. 328. Please clarify "as the pixel size of TROPOMI is high".
    - Changed: Removed in the re-write. Here intent was to understand the changes in emission with change in resolution. And in that context it is referred as large pixel sizes compaed 1km pixels.

47. 339. Table 2: please include percent differences.
    - Changed: Also it is now Table 3.

48. 351. How universal is this method? It seems to have many steps requiring thresholds which seem to have been selected based on specific examples. Would the same thresholds result in the desired results if the method was applied to other regions, other time periods?
    - Changed: APE generalize well as demonstrated for the Australia and Siberia case. See also different periods of the ensembles. See Sec 2 for filters and Sec 3 for emission estimates.
49. 355. Please clarify: 97.9% of what? Many plumes were rejected based on proximity to other plumes, lack of injection height data, … Such high percentage seems off.
    - Changed: See Item 40. We now speak of confidence level of the detected plumes.

**Typos, Minor/Grammatical comments**
50. line 5: please explain "APE"
    - Changed.
51. 7: please explain "VIIRS"
    - Changed.
52. 10: "IS4FIRES"?
    - Changed.
53. 15: should month names be spelled without abbreviation?
    - No. Changed accordingly.
54. 30: "CO emissions due fossil fuel burning has been on increase", consider rewording to "CO emissions due to fossil fuel burning have increased"
    - Done.
55. 32-33: please consider rewording "Thus, it becomes essential to understand the effect of CO on air-quality and climate by measuring it accurately on a global and local scales which helps to quantify CO emissions" for readability
    - Done.
56. 65: please correct "using the the wind"
    - Done.
57. 74: since the present tense was consistently used before (lines 71-74), please consider rewording to "The study results are deliberated in Section 3". Consider using "discussed" instead of "deliberated".
    - Done.
58. 80: "extracts TROPOMI CO data"
    - Done.
59. 87-88: please explain acronyms as they are introduced
    - Changed in the whole manuscript.
60. 90: Should "Furthermore" be "From now on" or similar?
    - Changed.
61. 92: also 5.5 x 7 km^2, since all but one of the cases analyzed here postdate August 2019.
    - True. Now included in manuscript.
62. 95: "low density areas"?
    - Yes, changed.
63. 104: "for part of data granule S5P_OFFL_L2__CO_____??? over Australia"
    - We have added Orbit number in the manuscript. It is also a unique identifier along with the product version. Line 109.
64. Fig. 4: the map shown in panel (b) seems to display zones of maximum slope change or, as the text states, zones of "gradient" change. However, panel (b) is labeled "Elevation map", which does not seem very appropriate. Please reword.
    - Reworded in the manuscript.
65. Changed 125: Ielev does not show a continuous variable (like elevation, or CO value) but it rather shows where the maximum change in that variable occurs. Consider renaming it to Iedge or similar.
    - Changed : To I_grad in manuscript. See line 137.
66. 149: Where does "14" come from? Eq. 3 is not needed, since it does not add to what's already in the text.
    - Changed: Removed.
67. 164. Please provide reference for the cross-sectional flux method.
    - Changed: Added to manuscript. See line 183.
68. Fig. 5 caption: please correct typo, "transaction" should probably be "transect"
    - Changed.
69. 197, 198. "remove overlapping fires". Fires like those shown in Fig, 6 are not overlapping, both the fire sources and the plumes appear quite distinct. Consider rewording "overlapping" by "closer than … km".
    - No changes. See item 34.

70. 220. does "on the right-hand side" refer to equation 8? If yes, then consider rewording to "The velocity v" or similar. If not, please explain what does it refer to.
    - Changed as suggested and referred to the equation (see line 242).
71. 244. Please clarify if "the velocity at the TROPOMI measurement time" refers to wind velocity.
    - Changed: Yes. Added to manuscript Line 261.
72. 265 Is "w.r.t" acceptable in a manuscript?
    - No. Changed.
73. Fig. 9. The blue crosses and blue dots are too similar to tell them apart. Consider using other symbols or separate colors instead.
    - Changed.
74. 332. "TROPOMI"
    - Changed.
75. 363. 22 out of how many cases? Alternatively, please provide a percent value. Otherwise, "22" alone is not informative.
    - Changed.

---

## Author Response (AR1)

Dear Prof. Dr. Christoph Knote,                                                15/03/2023

Thank you for giving us another opportunity to submit a revised version of our manuscript titled *Plume detection and estimate emissions for biomass burning plumes from TROPOMI Carbon monoxide observations using APE* v1.1 to EGU. We appreciate the time and effort that you have dedicated to providing your valuable feedback on our manuscript. We are submitting the revised manuscript with few major revisions based on the comments and suggestions from referees. Sections 2, 3 and 4 have been re-written and we have added Appendix B to the manuscript.

Along with these major revisions, we have
1. Changed changed the software version to APE 1.1. Changes in software are
    1. We have shortened plume length parameter filter from 40km to 25km to facilitate the identification of more plumes.
    2. We have done some bug fixes and re-wrote some parts of the code. These do not influence the results.
2. We have submitted the reply to each referee. However, I do now know if the reply to referees needs to be attached to this document.

We look forward to hearing from you in due time regarding our submission and to respond to any further questions and comments you may have.

Sincerely,
Manu Goudar
SRON Netherlands Institute for Space Research, Leiden,
The Netherlands

---

## Author Response (AR2)

Dear Prof. Dr. Christoph Knote,                                     21/07/2023

Thank you for giving us another opportunity to submit a revised version of our manuscript titled *Plume detection and estimate emissions for biomass burning plumes from TROPOMI Carbon monoxide observations using APE* v1.1 to EGU. We appreciate the time and effort you have dedicated to providing valuable feedback on our manuscript.

We are submitting the revised manuscript with a major revision based on the comments and suggestions from the second referee. Again, the manuscript has been rewritten to make things clearer. No changes have been made to the software or the data that was previously reported. The only new addition to the manuscript is section 3.2.2 on ERA5 uncertainties as suggested by the referee.

The detailed reply to Referee 2 can be found on the following pages.

We look forward to hearing from you in due time regarding our submission and to respond to any further questions and comments you may have.

Sincerely,
Manu Goudar
SRON Netherlands Institute for Space Research, Leiden,
The Netherlands

Dear Referee,

We would like to thank you for the constructive comments and recommendations on our manuscript. We have done several changes to the manuscript. Our replies to the comments are given below. The original comments are numbered, given in black and the answers are given in blue. The adjustments in the revised manuscript are specified in detail below where the page and line numbers refer to the revised manuscript.

1. The results presented show that this algorithm 1) can find plumes from approximately 4% of the fire clusters identified (where fire clusters are groups of 10 or more fire counts detected by VIIRS) and 2) produce emission estimates for that 4% of cases. That 4% value is never disclosed in the text. An unexplained 97.7% value is provided instead; more on this below.
    a. The Algorithm Application section states "our analysis confirms the applicability of our algorithm to other areas with a confidence of 97.7% of the cases." This statement is unclear and not supported by any explanation in the text. From the Conclusions section: "APE can reliably detect and estimate emissions automatically for 97.7% of the cases." This one is misleading, plain and simply. This issue was already identified in the first review; no satisfactory action was taken by the authors.
    b. The authors have not explained where the 97.7% value comes from, neither in the two versions of the manuscript nor in their responses to the first review. This reviewer's guess follows.
    c. (From Table 1) 221 "visual inspection" cases are approximately 97.7% of 226 "emission estimation" cases. The 221 "visual inspection" cases represent less that 4% of the 5562 "fire clusters" identified in VIIRS data.

Changed:

To quote a data yield of 4% does not adequately describe the situation. To evaluate the APE performance, the mission and observation aspects of TROPOMI must be separated from the algorithm.

1. TROPOMI data with a plume signature were only available for 16% of VIIRS counts. This is a data feature and not an algorithm feature and can be attributed to cloud coverage and the detection limit of TROPOMI. We are completely transparent about this in the manuscript in "Data preparation" section and the "Algorithm application" section. We believe saying "we only detected 4% of the fire counts" leads to misrepresentation of the algorithm.

2. "The 16% of VIIRS counts have TROPOMI plume signature" is highlighted in the manuscript in lines 12, 419-422.

3. Furthermore, we have reworded and made it clearer what 97.7% represents. We highlight that 97.7% is the true positive confidence in the APE's output. This shows that the output of APE can be trusted with 97.7% certainty.

4. Changes related to the above statements have been made in several places in the manuscript. Specifically, see lines 12-14, 284-292, 419-422.

2. The resulting emission estimates are not compared to any measurements; thus their validity remains unknown. Negative emission values are questioned and deemed invalid, as they should. Positive emission values are not questioned.

   Changed: To the best of our knowledge, no independent, high-quality estimate of CO point-source emission is currently available. Therefore, directly comparing the emissions with independent data is difficult. We discuss this in the paper referring to Sherwin et al. (2023) who validated satellite $CH_4$ data using controlled emission releases of point sources of methane for detection and quantification. No such validation can be done on CO.  However, Rowe et al., (2022) did show that the integrals of TROPOPMI CO data along the plume transects were ≈7.2% higher than the aircraft measurements after corrections for a few fires in the US. However, they do not report emissions. Thus, direct comparisons are difficult. We do add a paragraph about this. See lines 333-339.

3. The revised version still has language issues such as mismatched subjects and verbs, missing prepositions, abbreviations in the main text, typos, etc.

   Changed. The manuscript has been checked by a native speaker.

4. The revised text contains repetitions, e.g., the narrative on the number of cases appears in Abstract, Algorithm Application, and Conclusions, as well as in Table 1.

   Changed: Complete narrations of the cases in different sections have been changed in the manuscript. The complete analysis can only be found in Section 3.2, and Conclusions and Abstract briefly summarize the same.

5. The manuscript states that errors introduced by the wind data cannot be calculated due to lack of wind uncertainty information. Please note that the ERA5 ensemble data (members, mean, and spread; provided with the ERA5 data) describe uncertainties in the observations.

   Changed: Following the editor's suggestion, a new section has been added on ERA5 random uncertainties. However, we note in the manuscript that the ERA5 ensemble data only approximate random uncertainties and so, no information on the systematic errors or biases can be derived (see Section 3.2.2, see also (https://confluence.ecmwf.int/display/CKB/ERA5%3A+uncertainty+estimation).

Sincerely,
Manu Goudar,
SRON Netherlands Institute for Space Research, Leiden,
The Netherlands

---

## Author Response (AR3)

Dear Prof. Dr. Christoph Knote,                                    26/07/2023

We authors wholeheartedly thank you for accepting the revised version of our manuscript titled *Plume detection and estimate emissions for biomass burning plumes from TROPOMI Carbon monoxide observations using APE* v1.1 for publication in EGU.

We are submitting the final manuscript with a few minor format related corrections in dates, units, references, on referring to section and figures to adhere to the manuscript guidelines (for eg. use of Sect. instead of Sec. for section). Two changes are made, first one is the correction of date and time in Figure 5. This has no effect on paper. Second one is the name of one of the co-authors *Juliette Anema* has been changed to *Juliette C.S. Anema.*

Sincerely,
Manu Goudar
SRON Netherlands Institute for Space Research, Leiden,
The Netherlands